# Land Consumption for Current Diets Compared with That for the Planetary Health Diet—How Many People Can Our Land Feed?

Anna-Mara Schön * and Marita Böhringer

Department of Business, Fulda University of Applied Sciences, 60549 Frankfurt, Germany; marita.boehringer@w.hs-fulda.de
* Correspondence: anna-mara.schoen@w.hs-fulda.de

**Abstract:** Nowadays dietary habits in many countries are disconnected from the locally available resources and land. Current diets harm ecosystems and people's health. (Re-)regionalising food systems and aligning diets with planetary boundaries can be one way to reconnect people to the food that they eat. Within academic discourse, there are numerous debates about the benefits and drawbacks of regional agriculture, and the circumstances that determine the viability of regional agriculture as a preferable approach. An argument that often merges is that feeding a whole population using local resources cannot be accomplished. However, is this true? To test this argument, we used statistical data and created a framework to calculate land consumption in square meters per capita according to different dietary habits, among other factors. This study will focus on scenario analyses for the region of Hesse, Germany—with an emphasis on the livestock sector—as land consumption for the production of meat, milk and eggs is relatively high there. Our results show that the region is far from being able to feed the current livestock population and that it does not have the land to support the livestock needed to sustain current consumption patterns. However, the region could support a smaller livestock population with the implementation of the planetary health diet, especially if farmers were to adopt crop rotation systems and (more) extensive husbandry.

**Keywords:** self-sufficiency degree; planetary health diet; land consumption; food sovereignty; livestock; consumption

## 1. Introduction

Food production accounts for about one-third of the total greenhouse-gas emissions caused by human activity [1]. Approximately 20% of these are due to food transport [2]. Globally, one out of three people is overweight or obese, whereas one out of nine people is under- or malnourished [3]. In Germany, about 6 million children receive lunch at school or daycare centres on a regular basis, yet one in six children is overweight or obese [4]. This contradiction shows that the current food system is not working properly; consequently, both agriculture and consumption behaviour must be reconsidered.

Short food supply chains have been shown to have a positive impact on health [5] (e.g., ancient varieties and landraces, as they are often more nutritious, are typically more cultivated and sought after in short food supply chains) [6], climate (e.g., preserving agrobiodiversity and reducing $CO_2$ emissions from transportation) [7,8] and the local economy (e.g., regional value and job creation [9], and lower transportation costs) [8,10]. Even before the COVID-19 pandemic and the Russian invasion of Ukraine— two events that have unequivocally proved the risks of dependency on global supply chains [11,12]—many scientists, politicians and civilians agreed that food systems needed to not only become more resource- and climate friendly, but also more regional, not least to be able to guarantee national (food) sovereignty [13–15]. Further, regionalised agriculture can help promote sustainable agricultural practices, as it is easier for consumers and policy makers to track

and monitor the origin, quality and safety of food this way [16–19]. As the supply chains and transportation are shorter in regionalised agriculture, the products are fresher and of higher quality, and they require fewer preservatives or other methods that extend shelf life [19,20]. For instance, one study found that kitchens that had switched to regional supply could reduce food waste by 20% [21].

Sustainable food practices and food sovereignty are also demanded by the global peasant movement, La Via Campesina, which is supported by actors of nongovernmental organisations, academia, and the peasant community, all of whom strive for more sustainability, resilience and food sovereignty in their communities and regions [15]. Key factors in food sovereignty include prioritising local agricultural production; peasants and landless people's access to land, water, seeds and credit; the right of farmers and peasants to produce food; the right of consumers to decide what they consume and how and by whom food is produced; the right of countries to protect themselves against too-low-priced agricultural and food imports; the right to impose taxes on excessively cheap imports if they commit themselves in favour of sustainable farm production; and the recognition of the right of female farmers, who play a major role in agricultural production and food systems [22].

The main goal of food sovereignty is to prioritise people's nutrition rather than neoliberal policies and international trade [22], and to transform the current food system from the industrialised system back into peasantry farming. The current industrialised food system can be traced back to a few historical events, i.e., the first (1870–1930s) and second (1950–1970s) food regimes, the Green Revolution (from the end of the Second World War until the late 1970s) and the industrialisation of agriculture and trade liberalisation, all of which played their part in producing and advertising cheap food, such as corn, rice and wheat, but also in achieving the mass-production of animal products to fuel cheap labour and strengthen the hegemonic role of the United States and capitalism [15,23,24].

Apart from the negative impact on people's health, the increasing demand for and dependence on agricultural imports and the overall level of poverty, the Green Revolution and agro-industrialisation have significantly negative impacts on our climate and biodiversity [15,22,25]. For instance, while the production of corn per acre increased by ~2.4 times from 1945 to 1970, fuel inputs rose by ~3.1 times. As early as 1973, scholars discovered that 80 gallons (897 litres per 1 ha) of gasoline are consumed per 1 acre of corn produced. Such examples showcase why greenhouse-gas emissions from food production are so high nowadays [1].

With this in mind, to (re-)gain food sovereignty, in many parts of the world, food policy councils and other food activists, such as the transition town movement, the slow food movement, etc., have been established [26–28]. They demand independence from global food supply chains and resilience, and that regions, whether small or large, be able to feed themselves according to the available arable land and pastureland.

On the other hand, arguments against short and more sustainable food supply chains persist. Advocates for the status quo argue that crops should be grown where the highest yields can be achieved and that transport costs and emissions are only a small part of the total cost, thus hardly playing a role [29,30]. Low yields are not a viable solution, according to this argument, because population growth, income growth and changing diets are predicted to increase the demand for agricultural products by about 60% by 2050 (from the base year of 2005) [31,32]. The increase in animal welfare in Germany and, thus, the reduction in the number of animals kept for meat production demanded by more sustainably minded people are in contrast to the argument that meat should not become a luxury good that only wealthy people can afford [33–35].

The lobby that promotes this argument is strong and successful in averting any significant (fast) change toward greater climate protection and nature conservation.

A variety of important studies have explored our global food system and its impact on the environment, biodiversity, climate and health, focusing on global data [36–40].

In order to calculate the availability of food and to provide a basis for agricultural and food policy, a number of models that are dedicated to this topic with different focuses already exist. These include, for example, FAO models such as the partial equilibrium and computable general equilibrium for forecasts, Food Security Models for assessing food security and Food Balance Sheets, which provide information on imports, exports and production at the national level [41–43]. However, these models have faced criticism for being inaccurate, incomplete and neglecting the ecological and social aspects of agriculture [32,44–48]. To address this criticism and take regional specifics into consideration, our calculations are based on regional data from the state of Hesse. With this approach, we aim to make agricultural production data understandable and tangible at the regional level and obtain more concrete measures that fit local conditions. Our scenarios go beyond pure model calculations and show what would change if ecological agricultural practices were integrated and consumers changed their dietary habits.

Additionally, despite scientific reports and models repeatedly showing the threat of climate change and the overuse of the Earth's resources [49], people act too slowly or do not react at all to the threats that we are facing. What is the reason for this? Studies modelling global data do not seem to convince people to act locally, possibly because these data are too abstract to understand and foreign to people's own experiences (proximity effect) [50]. Psychology shows that more information may not be the key to climate action and that doomsday messages tend to fail [51]. The use of status, metrics and friendly competition works better: "Carbon footprints have been useful because people can improve. You can actually have a positive trajectory and feel good about that. Then they can compete. Everybody likes to have that smiley face; no one likes to have that frowny face" [51].

With this in mind, the aim of this study is to provide people with a tool to calculate their land consumption in square meters per capita, depending on the way they eat, and to determine the impact on farmers' practices. With our metrics at hand, people who change their dietary habits can determine their own decrease in land consumption, awarding themselves that "smiley face". Studying smaller areas, but in detail, is essential [32].

To calculate and process details, local conditions must be considered, such as the current local supply and demand, types of agriculture (e.g., pastureland vs. arable land), soil conditions, etc. Even though our approach focuses on a small area, the state of Hesse in Germany, we propose that our findings, especially the approximate square metres required by the average local person to sustain their diet, be generalised to at least a wide range of European regions and probably most areas of the Global North, particularly because the way that food consumption in Central Germany is similar to that in many other regions [52].

Considering all these arguments for and against a local and climate-friendly food system, a range of research questions and assumptions emerge:

RQ 1: How many animals would be necessary to satisfy the demand for regional animal products (meat, dairy and eggs) in the context of current consumption patterns?

RQ 2: Can local land feed the number of currently kept livestock and the number of livestock necessary to achieve a 100% degree of self-sufficiency (SSD)?

RQ 3: How much food is grown on local fields, and how much would be necessary to feed the local population in a plant-based, healthy, appealing and diverse way, as recommended by the planetary health diet (PHD)?

To this point, these questions have not been answered, at least not in detail. We assumed that the studied area is not big enough to feed the animals needed for ood—including meat, eggs and dairy products—with local resources, and that the crops were not diverse enough to provide healthy food for everyone. Thus, we raised the following question:

RQ 4: How would land consumption change if everyone consumed in a more environmentally friendly manner—specifically, consuming fewer meat, eggs and dairy products, as, for instance, proposed in the planetary health diet by the EAT-Lancet Commission in 2019 [13]—and agriculture became more sustainable and environmentally friendly?

To answer these questions, we studied six different regions within the state of Hesse, and the whole state of Hesse, in four scenarios. The first scenario considers current consumption patterns, which are based on current cultivation statistics and the current number of livestock (referred to as *current* in the following). The calculated self-sufficiency degrees provide insights about the supply situations in these regions. To answer RQ 1, the second scenario calculates the livestock necessary to meet the current consumption patterns—to reach a self-sufficiency degree of 100% (referred to as *SSD 100%* in the following) [53–56], and provide answers to RQ 2. The third (*PHD*) and fourth (*extensive*) scenarios are based on the consumption recommendations of the planetary health diet and thus analyse the livestock necessary for both the adapted consumption patterns (referred to as *PHD* in the following) and for changing the type of cultivation to a seven-year crop rotation system and extensive husbandry (referred to as *extensive* in the following). Scenario *PHD* answers RQ 3, while scenario *extensive* sheds light on RQ 4.

## 2. Method and Concept of the Study

A detailed framework was established to calculate the land consumption in $m^2$/capita and the different levels of self-sufficiency. Figure 1 displays an overview of the framework development. To answer the research questions RQ 1–3, we first identified the areas to be studied (step 1, cf. *2.1 Regions under Consideration*). We then looked at specific agricultural statistics and current dietary habits and the planetary health diet (steps 2 and 3, cf. *2.2 Selected Food Groups*). To estimate how many animals must be fed with what amount of fodder per year, we defined fodder examples based on the literature and expert interviews (step 4) and the so-called herd factors, stable place and slaughter quotas (step 5, cf. *2.3 Animal Production Rates and 2.4. Land Consumption for Animal Feed*). Then, we excluded the plants used for energy production from the total and considered further assumptions (steps 6 and 7, cf. *2.5 Plants for Energy Production* and *2.6 Further Assumptions*). Equipped with this information, we were able to consider different scenarios (steps 8–12, cf. *2.7 The Calculated Scenarios: Current, SSD 100%, PHD and Extensive*).

### 2.1. Regions under Consideration

We specifically analysed the whole German state of Hesse to calculate the self-sufficiency degrees and square meters per capita. Located within Hesse are (1) three governmental districts (Darmstadt (2), Gießen (3) and Kassel (4)), two smaller regions (the metropolitan area of Frankfurt/Main, including all the bordering counties (5) and the rural county of Marburg-Biedenkopf (6)), which are shown in detail below (Figure 2) [57].

### 2.2. Selected Food Groups

We selected and adapted the food groups used in the planetary health diet and combined them with current consumption patterns and the consumption recommended by the planetary health diet (PHD) per capita (Table 1). Since the PHD recommends 2500 kcal per person per day, we adjusted this figure to 2150 kcal (86%), i.e., the calculated median across the age groups of the Hessian population and the respective quantities required.

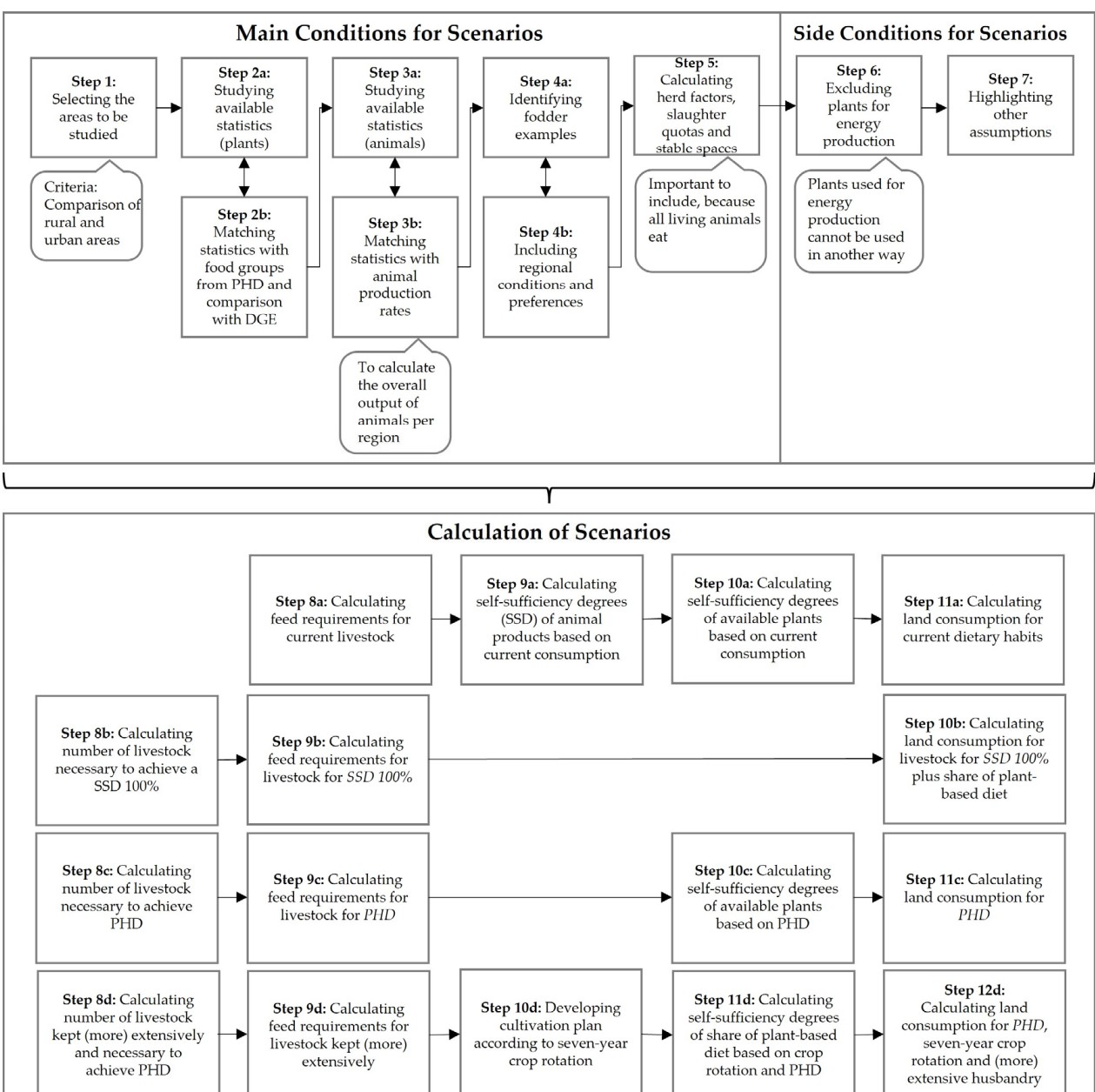

**Figure 1.** Overview of Framework Development. The upper part of the figure shows the step-by-step procedure of the study and the conditions taken under consideration, separated into main conditions (step 1 to 5) and side conditions (step 6 and 7) and aiming to provide answers to research questions 1 to 4. Based on the defined conditions, the lower part of the figure presents the procedure for calculating the four scenarios: calculating land consumption for current dietary habits and current livestock (steps 8a–11a, *current*); calculating land consumption for a self-sufficiency degree (SSD) of 100% based on current dietary habits (steps 8b–10b, *SSD 100%*); calculating land consumption for a diet based on the planetary health diet (PHD) and comparison with the Deutsche Gesellschaft für Ernährung (DGE; steps 8c–11c, *PHD*); calculating land consumption for PHD in combination with a more sustainable type of agriculture, namely, seven-year crop rotation and (more) extensive husbandry (steps 8d–12d, *extensive*).

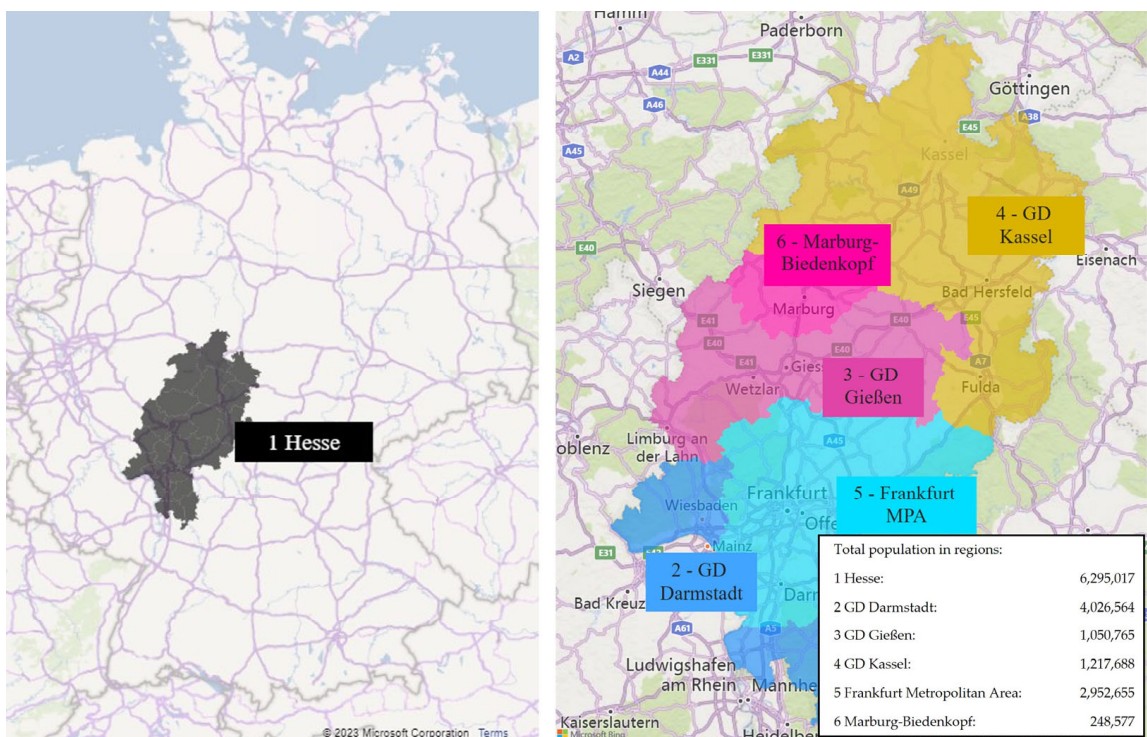

**Figure 2.** Location of the Analysed Regions and their Total Population. The left side shows region 1, the whole state of Hesse, which is located in the middle of Germany, and the right side shows where in Hesse the regions 2–6 are located. Regions 2–4 are the governmental districts (GD) of Darmstadt (2), Gießen (3) and Kassel (4); region 5 is the Frankfurt metropolitan area (F MPA), consisting of the city of Frankfurt/Main and bordering counties, and part of region 2; and region 6 is a more rural area, the county of Marburg-Biedenkopf (M-B), about 80 km north of Frankfurt, and part of region 3.

**Table 1.** Consumption in kg/capita p.a. In the first column on the left, the different food groups based on the planetary health diet are displayed. In the second column, the current consumption based on the statistical data is shown. The third column shows the annualised PHD recommendation in kg. The fourth column downscales this recommendation to a daily intake of 2150 kcal/capita, i.e., the calculated Hessian median in relation to the age groups living in the region and the calorie intake recommended by the German Nutrition Society (DGE—Deutsche Gesellschaft für Ernährung) per day and age group [13,58–62].

| Food Group | Current Consumption [1] | Consumption Recommended by PHD 2500 kcal | Consumption Recommended by PHD 2150 kcal |
|---|---|---|---|
| Cereals | 85.4 | 84.7 | 72.8 |
| Legumes | 0.9 | 27.4 | 23.5 |
| Potatoes | 71.7 | 18.3 | 15.7 |
| Vegetables | 98.6 | 109.5 | 94.2 |
| Fruits | 66.5 | 73.0 | 62.8 |
| Plant-oil | 14.5 | 18.9 | 16.3 |
| Nuts | 5.0 | 18.3 | 15.7 |
| Sugar | 33.6 | 11.3 | 9.7 |
| Milk, equiv. diary | 409.6 | 91.3 | 78.5 |
| Eggs (pcs.) | 239.0 | 75.3 | 64.8 |
| Red meat | 42.0 | 5.1 | 4.4 |
| White meat | 13.1 | 10.6 | 9.1 |
| Fish | 14.1 | 10.2 | 8.8 |

[1] All data are expressed in kg per capita per year (except eggs).

We emphasised the PHD food groups, which are also included in the German Nutrition Society's (DGE) recommendations and for which a solid data base is provided. These groups include grains, grain products and potatoes, which are important sources of energy, carbohydrates and dietary fibre [63,64]. The next food groups comprise vegetables, salad and legumes (such as peas, beans and lentils). Vegetables are rich in vitamins, minerals, dietary fibre and phytochemicals, and legumes are a good source of protein and dietary fibre and constitute a good meat alternative. Fruits are rich in vitamins, minerals, fibre and phytochemicals; nuts and seeds are important sources of nutrients and therefore part of a healthy diet. We did not include nuts in our study, as they are hardly cultivated in Hesse. There are no exact statistics on fruits. The region mainly cultivates strawberries, cherries and apples. Local apples are used mainly to produce juice and cider. As these products grow only in certain areas of Hesse and are sold mostly by means of direct marketing, we decided not to include them. Milk and dairy are a good source of calcium and provide high-quality protein, iodine and vitamins A, $B_2$ and $B_{12}$. Meat also provides high-quality protein, and vitamin $B_{12}$, selenium, zinc and iron. Processed meat is rich in saturated fatty acids and salt. White meat, as both the DGE and the PHD state, is preferable to red meat, as there is no relationship between the former and cancer according to current knowledge [63]. The DGE suggests a moderate consumption of red meat due to high greenhouse-gas emissions from ruminants, such as cattle, sheep and goats. The PHD proposes radically decreasing red meat intake compared with the current average intake because of health issues and the negative environmental impact [64]. Fish, another food group, is a high-quality protein source; fatty fish species are rich in valuable long-chain omega-3 fatty acids, and sea fish is high in iodine. We did not include fish in our study, as it is hardly cultivated in Hesse. Eggs are a good source of protein and fat-soluble vitamins, though the yolk is high in fat and cholesterol. Current studies do not show an upper limit for egg consumption; however, the DGE does not recommend an unlimited amount in the context of a plant-based diet [63]. The PHD limits eggs to about one egg per week [64]. Rapeseed oil is recommended by the DGE as plant-oil, another food group, as it contains the lowest proportion of saturated fatty acids, a high content of monounsaturated and polyunsaturated fatty acids, vitamin E and a good ratio of omega-3 fatty acids to omega-6 fatty acids [65]. The PHD does not recommend any specific oil but highlights the importance of unsaturated fatty acids [64]. In our study, we calculated the amount of available plant-based fat based on rapeseeds and sunflowers because both of these plants are grown regionally.

The yields per hectare based on the German Federal Statistical Office [66] and crops can be found in Tables A1–A3, respectively, and the extrapolated consumption by region in kg and ha is shown in Tables A4 and A5. Yields may significantly vary, depending on the type of soil, the quality of seeds used and the weather conditions.

### 2.3. Animal Production Rates

Regarding animal products, to calculate the actual self-sufficiency degree, we used averaged conventional production rates, such as average milk yield per animal group (9358 kg/cow/year; average slaughter weights: 230 kg per cattle, 21 kg per sheep, 11 kg per goat and 98 kg per pig; chicken lying performance: 288 eggs/chicken/year; slaughter weights of poultry: 2 kg per broiler chicken, 5.2 kg per goose, 10 kg per turkey and 2.2 kg per duck).

The herd factors (how many animals need to be kept to maintain the herd at a constant level) and the slaughter rate (how many animals "occupy" a stable place per year before they are slaughtered) can be found in Table A6 [67–76]. These factors are important in calculating how much feed is eaten in total per year and not per animal. For extensive husbandry, some of these figures were adapted, such as eggs (180 instead of 288) (cf. Table A6).

### 2.4. Land Consumption for Animal Feed

To calculate how much fodder the region can provide for which number and type of animals (permanent pastureland and arable land), we had to define exemplary fodder rations based on the given literature and expert interviews [67,68,77–79] (cf. Tables S1 and S2). Our fodder rations are based on regional cultivation practices, such as high rations of maize silage and cereals and lower rations of legumes and lucernes. Based on our conversations with farmers, we are aware that these rations vary greatly depending on the farm and region and that it is impossible to obtain realistic data based on a few feeding examples (each farmer has his/her own feeding practices). However, this step was necessary in order not to make the calculation model too complicated or chaotic.

### 2.5. Plants for Energy Production

We drew on statistics provided by the state of Hesse regarding arable land (ha and cultivated crops), pastureland, permanent crops, and types and number of animals. These statistics can be found in Table S3 [80].

The statistics indicate crops per ha, not yield per region. The statistics do not reveal which crops are used for human consumption, fodder or energy production. However, we can infer that about 95% of legumes grown are for feeding [81], as are—in total—triticale, lucerne, corn-cob mix and maize silage (about 20% of the total cropland). We estimated that in Hesse, at least 2.1% of wheat, 9% of silage maize, 1.84% of sugar beets, 0.4% of potatoes, 7.2% of rapeseed and 1% of legumes are used for energy production or industrial purposes [82,83]. Table A7 shows the results. Table S4 shows the calculation basis. These percentages were subtracted from the total amount per food group and were not considered in future calculations.

### 2.6. Further Assumptions

Approximately 30% of the total production of wheat and 55% of rapeseed do not enter the market for human consumption, because there are remaining shares after threshing, milling and oil-pressing processes that can be fed to animals as protein-rich cake or meal. Despite knowing that the areas studied do not have large processing plants—such as oil mills, grain mills, threshing crop processing or sugar factories—we still considered that all harvests are instead processed and consumed here by humans and animals, in order to calculate the local self-sufficiency degrees. We excluded any kind of food waste, although we are aware of the extent and problematic nature of this loss. One reason for this is that we estimated that if supply chains were localised and made more sustainable, a circuit economy would be built, in turn drastically reducing food waste (the leftovers would be fed to animals, and the products would be valued more and used when available).

### 2.7. The Calculated Scenarios: Current, SSD 100%, PHD and Extensive

Based on these data, we were able to model the following four scenarios, focusing on livestock before calculating how much land is left for the plant-based share of the diet:

- livestock for *current*: The necessary feed requirements for current livestock in ha (cf. Table S5) in the different regions and the self-sufficiency degrees of red meat, white meat, eggs and milk/dairy products;
- livestock for *SSD 100%:* The number of livestock, including herd factors/stable places, necessary to ensure our current consumption patterns (100% self-sufficiency degrees of animal products; cf. Table S9);
- livestock for *PHD*: The number of livestock, including herd factors/stable places, necessary to ensure the consumption level recommended by the PHD (100% self-sufficiency degrees of animal products; cf. Table S10);
- livestock for *extensive*—livestock, including seven-year crop rotation system: The self-sufficiency degrees of plants and animal products in the utopian case in which all farmers use a seven-year crop rotation system and only keep animals extensively instead of intensively (as it is mainly practiced today) (cf. Table S11).

Further, we calculated the self-sufficiency degrees of plants for human consumption (cereals (excluding triticale), sugar, potatoes, oil from rapeseeds, legumes and vegetables) based on the current consumption patterns and the PHD recommendations (cf. Table S6). We did not focus on oil from sunflower, as it only plays a minor role in local farming practices. For each scenario, we calculated the approximate total and per capita necessary land consumption of pastureland and cropland.

The *current* and *SSD 100%* scenarios aimed to answer RQ 1 (How many animals would be necessary to satisfy the demand for regional animal products (meat, dairy and eggs) in the context of current consumption patterns?) and RQ 2 (Can local land feed the currently kept livestock and the number of livestock necessary for a 100% self-sufficiency degree (SSD)?). To answer the first part of RQ 3 (How much food is grown on local fields?), we subtracted the arable land necessary to raise livestock from the total to reveal how much land is left for the plant-based share of the diet. By calculating the self-sufficiency degree, we could answer the second part of RQ 3 (How much would be necessary to feed the local population in a plant-based, healthy, appealing and diverse way, e.g., as recommended by the planetary health diet (PHD)?). With the results obtained so far, we confirmed our assumption that the area studied is not large enough to feed the animals that are needed for food—including eggs and dairy products—using local resources. By recalculating the land consumption for the necessary livestock based on the planetary health diet (*PHD* scenario), we mathematically decreased the land consumption per capita and proved that current agricultural practices were sustainable in general, albeit not in line with La Via Campesina nor diverse enough to provide healthy food to everyone (exceptions excluded). This led us to analyse RQ 4 (How would land consumption change if everyone consumed in a more environmentally friendly manner, specifically, less meat, eggs and dairy products, for instance, proposed in the planetary health diet, while at the same time agriculture became more sustainable and environmentally friendly?) The latter scenario is in line with the literature calling to include sustainable agriculture [32,47].

## 3. Analysis of the Status Quo: Low Self-Sufficiency Degrees—What Must Change?

Before presenting the results in detail, we provide a brief overview of the region.

### 3.1. Overview of the State of Hesse

Hesse is a state in Central Germany with almost 6.3 million inhabitants, 298 inhabitants per km$^2$, 302.53 billion EUR GDP (in 2021) and an unemployment rate of 4.9% (in 2022) [84]. In total, 15,128 farms can be found in this state, 688 of which are under 5 ha, and 536 of which are over 200 ha [53]. Most farms in this state are between 50 and 99 ha (3853 farms). Only 4241 of all the farm owners work as full-time farmers. A total of 10,221 farms keep livestock, mainly cattle (6429 farms keep 406,304 cattle, about one-third of which are dairy cows) and pigs (2407 farms keep 543,934 pigs). A total of 2108 of the farms work organically, of which 1674 keep livestock (63,006 livestock units) [53,69].

### 3.2. Arable Farming

In the state of Hesse, a total of 764,705 ha is used for agriculture (36% of the total area). Of this, 61% is used as cropland and 38% as pastureland, while 1% is used to grow permanent crops, such as fruits, nut trees and bushes.

Farmers cultivate vegetables on 8285 ha of land, and almost 32 ha is covered; farmers cultivate strawberries on nearly 1000 ha of land [55,56].

The shares of main crops cultivated on cropland—including organic crops—in each region are displayed in Table A1. The overall share of organic farmland in ha is between 10 and 21%, depending on the region; the share of organic vs. conventional farms varies between 14 and 20%. The state's goal is to increase the area of organic farming to 25% by 2025 (cf. Table S3) [85]. The area of organic farming per crop significantly varies with total cereals. In region 1 (Hesse), this area is 8%, of which wheat, spelt, Einkorn and corn maize/corn-cob mix represent 6% each; rye and triticale, 16% each; barley, 4%; oat, 27%;

other cereals, such as Emmer, millet, buckwheat and sorghum, 42%; silage maize, 3%; sugar beets, 2%; potatoes, 11%; rapeseeds, 0%; and legumes, 35%. The data show that the main crop cultures, such as cereals and maize, have a rather low share of organically produced crops, whereas niche products, such as Emmer, oat and buckwheat have a rather high organic share.

### 3.3. Current Production and Consumption of Animal Products

About 580,000 grazing animals—including close to 35,000 horses, roughly 544,000 fattening pigs, close to 33,000 breeding sows, 1.1 million broilers and other poultry and nearly 1.5 million laying hens—are currently kept in region 1 (cf. Table S5) [53,69].

The self-sufficiency degree of animal products varies quite significantly by region and product (cf. Figure 3 and Table S8, SSD animal products). The current production of animal products is the highest in region 4 (e.g., SSD of milk: 111%; SSD of red meat: 132%). Region 5 has most inhabitants, the lowest share of agriculture and, thus, low self-sufficiency degrees of animal products (e.g., SSD of white meat: 1.1%; SSD of eggs: 14%). Region 3 and region 6, which is part of region 3, raise cattle and dairy cows intensively, but still cannot meet the local demand (e.g., in region 3, SSD of milk: 64%; SSD of red meat: 54%). White meat and eggs are only produced to a considerable degree in region 4 (SSD of white meat: 125%; SSD of eggs: 55%).

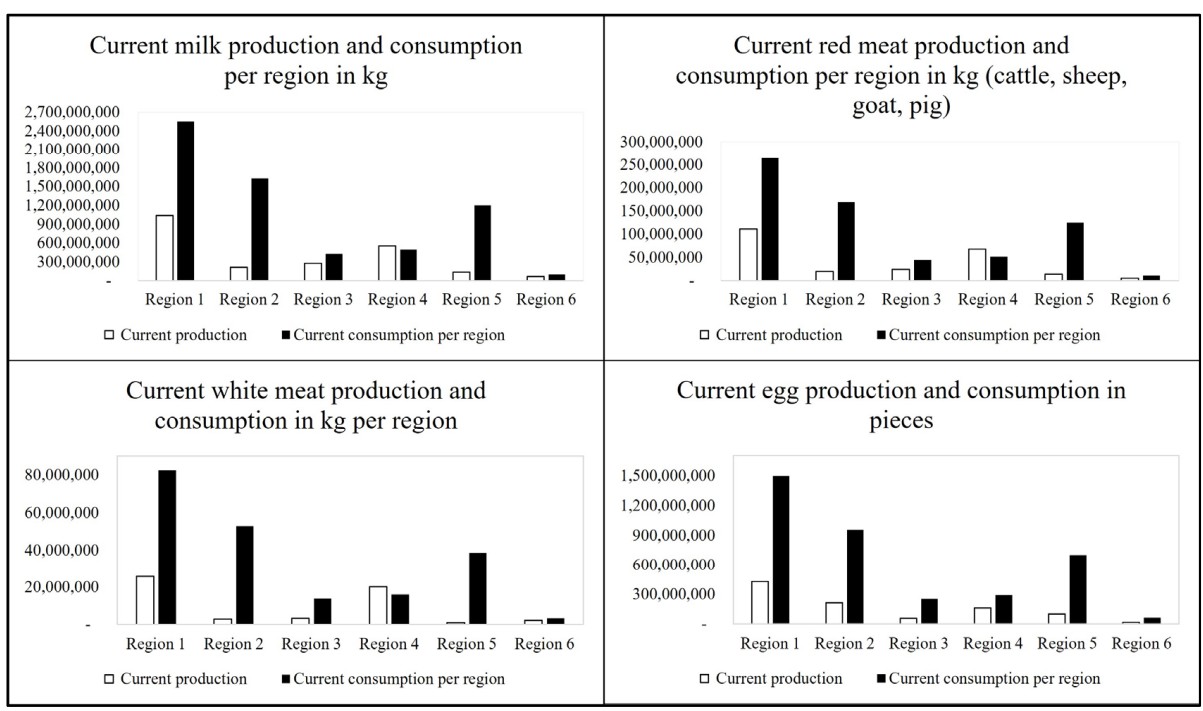

**Figure 3.** Current Animal Product Consumption by Region 1 to 6. Region 4 is an animal-producing region, with more milk and red and white meat being produced than consumed. In all areas other than region 4 consumption exceeds production for all food types.

### 3.4. Calculated Number of Animals for Current Consumption and Consumption According to PHD Recommendations

The next step of the analysis was to calculate the number of animals necessary to ensure the current level of consumption of animal products in each region (*SSD 100%*; cf. Table S9), to ensure the consumption recommended by the PHD (*PHD*; cf. Table S10) and to sustain the consumption recommended by the PHD with extensive animal husbandry and thus lower/slower output per animal (*extensive*; cf. Table S11). Figure 4 shows the results of region 1, the whole state of Hesse, and Figure A1 shows the results of all the regions. The herd factors/stable places reported in Table A6 were included in the calculation of

the results. The calculation of the livestock necessary for the different diets can be found in Table S12. In Hesse, for example, many more animals would be needed to meet local demand. If all the inhabitants were to eat as recommended by the planetary health diet, the farmers would have to keep far less livestock. If everyone in the region ate as recommended by the PHD and the livestock were kept extensively, the number of animals would increase to *PHD*, but only slightly. The results of the other regions are similar. Only in region 5, the most populated one, is the difference between the current and necessary livestock (*PHD*) less extreme because few animals live in the region. For poultry kept to produce meat, the differences are also less notable, because the PHD-recommended poultry levels are quite high. The results in Figure 4 answer RQ 1 (How many animals would be necessary to satisfy the demand for regional animal products (meat, dairy and eggs) in the context of current consumption patterns?). For instance, in region 1, currently 124,750 dairy cows and about 544,000 pigs are kept, though almost 363,800 dairy cows and about 1 million fattening pigs are necessary. If everyone ate by the PHD, region 1 would only need 61,000 dairy cows and 104,000 fattening pigs, which would increase to about 128,000 and 139,000 extensively kept dairy cows and pigs, respectively (cf. Table S5 and S9–S11).

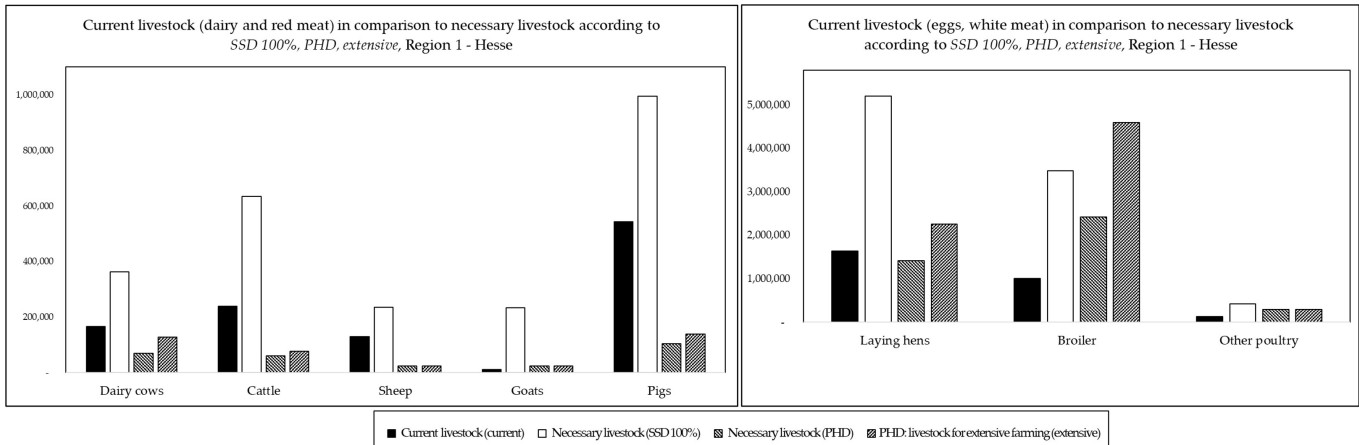

**Figure 4.** Current Livestock in Comparison with Necessary Livestock. The number of livestock needed in *SSD 100%* exceeds the number of currently kept livestock (*current*), whereas the number of livestock needed in *PHD* is much lower. If all the animals were kept extensively (*extensive*), the numbers would increase slightly to *PHD*.

### 3.5. Calculation of Feed Quantities and Land Consumption Required for Livestock

The next step based on the results in Figures 3 and 4 was to determine the amount of feed required for the livestock of the scenarios *current*, *SSD 100%* and *PHD* (for the *extensive* scenario, see Section 3.8). This step is necessary in assessing the land consumption for the current diet and for the planetary health diet.

Based on our fodder examples and yields (mainly from 2021 [66]), we calculated that the livestock currently kept in the different regions (1 to 6) would use between 62% (region 3) and 80% (region 4) of the available pastureland and between about 34% (region 5) and 75% (region 4) of the available crop-land if the total amount of fodder were produced locally (cf. Figure 5). Assuming an SSD of 100%, the necessary amount of feeding would exceed the regions' resources by up to 409% (region 5) for pastureland and by 256% (region 5) for cropland. On the other hand, based on the PHD consumption recommendations (i.e., the consumption of fewer animal products), the regions would only need between 14% (region 4) and 65% (region 5) of pastureland and between 14% (region 4) and 47% (region 2) of cropland, clearly indicating that extensive husbandry would be possible.

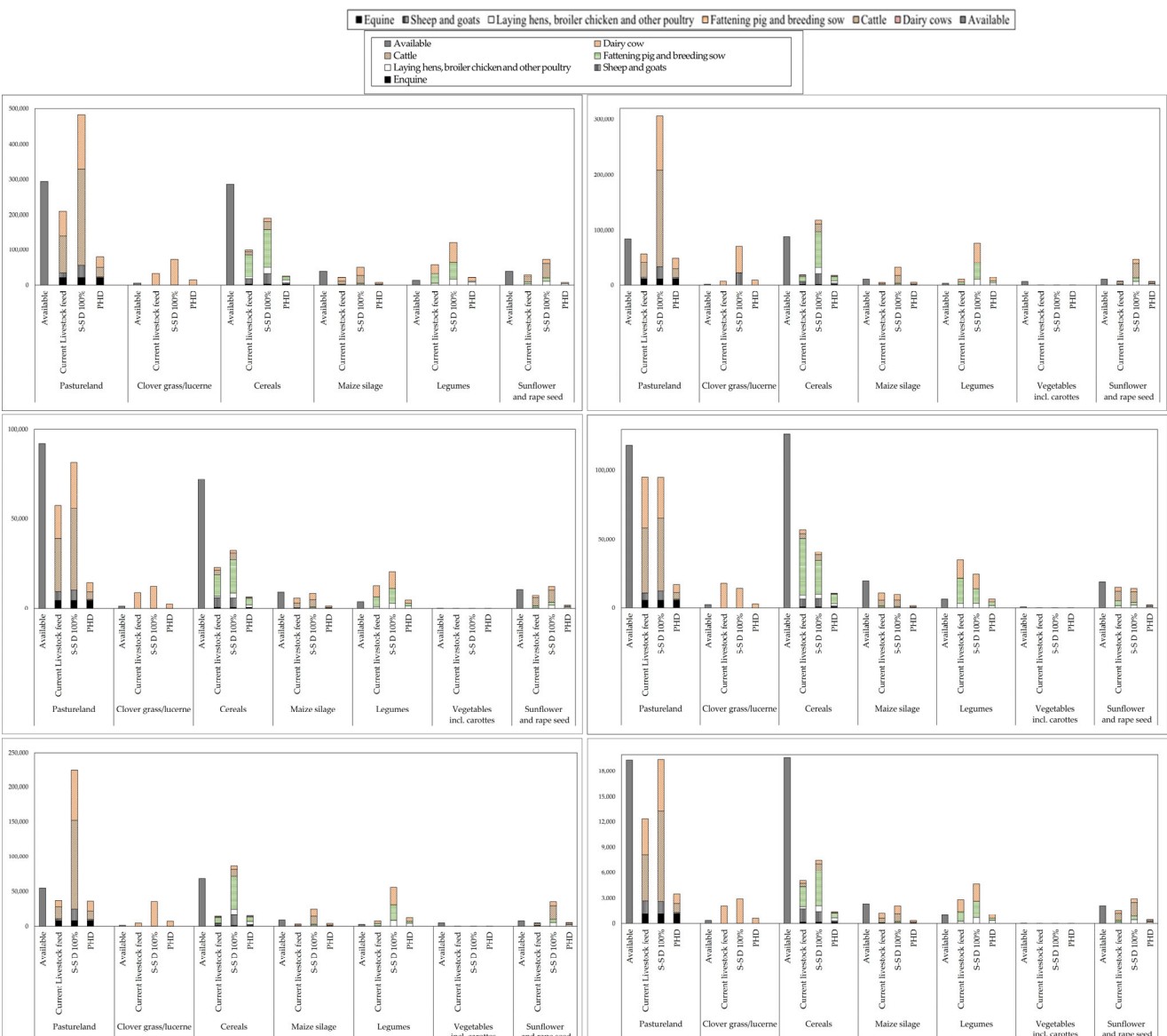

**Figure 5.** Land Consumption of Current Livestock in Comparison to Necessary Livestock, Regions 1 to 6. Each chart (regions 1–6 are shown from the top left (region 1) to the bottom right (region 6)) shows the amount of available pastureland and arable land (left bar); the amount that the current livestock needs for fodder (second left bar, *current*); the amount of fodder necessary to feed the livestock in *SSD 100%* (third left bar); and the amount of fodder necessary in *PHD* (right bar). The bars are composed of segments representing the demand of individual animal groups and are shown stacked (cf. Tables S2, S9–S11).

The main argument for keeping cattle (including dairy cows) is that ruminants are capable of processing non-edible grass, clover grass and lucerne into valuable protein for humans, although they emit a high share of methane as a result of their digestive processes [36]. Regarding the forage examples, pastureland does not seem to be used optimally, as out of the 294,288 ha of the available pastureland in region 1, only 208,985 ha was used in our calculation (Figure 5, left bars). Apart from the potentially underestimated feed rations in our forage examples, in these regions, it can be observed that many animals have to stay in the barn because direct grazing is not possible, requires too much effort or is simply difficult due to poor weather conditions. In addition, the proportion of concentrated feed is quite high in comparison to pasture forage for dairy cows and cattle, so milk and meat yields are high. Regarding regional grazing practices, hardly any hybrid grazing is

practised, which is corroborated by the low shares of goats and sheep in the total livestock (0.4% of goats and 1% of sheep). In the *SSD 100%* scenario, most regions would not have enough pastureland to feed all the animals (the necessary amount for region 1 is 482,996 ha), contrary to the *PHD* scenario. If everyone was to eat according to the PHD recommendations, there would be enough grazing land for all the animals that would then be needed (80,647 ha of pastureland).

The cultivation of lucerne, clover grass and legumes (status quo: mainly field beans and field peas for feeding livestock) on cropland is good for soil fertility, as nitrogen is bound and is a good source of protein for animals—not only for cattle and dairy cows, but also for pigs and poultry [86,87]. Especially organic farmers cultivating according to perennial crop rotation have clover grass silage and legumes in stock for their animals in the winter [68,81]. According to the statistics, the overall amount of lucerne and clover grass is only 1.14% of the total cultivated area in Hesse (cf. Table S7) [66]. The quantities grown are far too small to feed the current local livestock (available land in region 1: 5300 ha for lucerne and clover grass, and 13,277 ha for legumes). No region grows nearly enough of these protein crops needed for the current livestock (region 1: 33,823 ha for lucerne/clover grass and 58,312 ha for legumes), let alone the necessary livestock (region 1: 74,067 ha for lucerne/clover grass, and 120,601 ha for legumes). Not even the livestock of *PHD* could be kept with the current quantity of lucerne (region 1: 14,447 ha) and legumes (region 1: 21,522 ha). The small number of protein crops suggests that most animals are fed imported products (e.g., soybeans from Brazil).

Maize silage is a very important forage crop for cattle and dairy cows, as it can be used to increase the milk yield of dairy cows, among other applications. It is only produced in areas where it grows well. The amount of maize silage in Hesse—excluding usage for energy production—is about 40,000 ha. Although we adjusted our forage examples to the statistically grown amount of maize, the maize currently used to feed livestock amounts to 21,510 ha (region 1). Thus, we suspect that the share of maize grown for energy production is even higher than we assumed. The amount of maize silage would not be enough for *SSD 100%* in regions 1 (necessary: 51,272 ha), 2 (necessary: 32,796 ha; available: 10,636 ha) and 5 (necessary: 24,049; available: 8330 ha). The amount of cereals grown is enough for livestock for the *current* scenario in all the regions. It would also be enough for *SSD 100%* in region 1 (necessary: 191,130 ha; available: 286,368 ha), region 3 (necessary: 32,530 ha; available: 71,947 ha), region 4 (necessary: 40,662 ha; available: 126,916 ha) and region 6 (necessary: 7433 ha; available: 19,595 ha). The highest share in feeding is due to pig farming.

Oil-seed crops are also fed to animals, especially cattle; however, only a certain amount is fed to them, as the remaining share after oil pressing is high in proteins. Since we do not know the currently fed shares of complete oilseeds and oil press cakes, we calculated the share of oilseeds (rapeseeds and sunflower seeds) in whole seeds. In all the regions, the amount of currently grown oilseeds is enough to feed the current livestock, but not enough to feed the necessary livestock (except in region 4). For PHD consumption, the current crops would be sufficient. Tables S2, S9 and S10 show arable crops divided according to animal groups and aggregated according to the calculated necessary number of animals per diet (livestock feed of the scenarios *current*, *SSD 100%* and *PHD*, respectively).

The calculated self-sufficiency degrees provide evidence that the local land cannot feed the currently kept livestock nor the livestock necessary to achieve a 100% SSD (RQ 2).

*3.6. Direct Human Consumption—Self-Sufficiency Degree (SSD 100% and PHD)*

The necessary current consumption of plants amounts to 226,522 ha for region 1, which decreases slightly in the *PHD* scenario (region 1: 225,090 ha). By only considering plants currently grown for human consumption (wheat, spelt, Einkorn, rye, barley, oat, other cereals, sugar beets, potatoes, oilseeds and legumes) and considering that legumes comprising only 5% of the total amount—as 95% of legumes are grown for animal feeding and vegetables—the total available land is 328,829 ha (region 1). Table A8 shows the available ha currently used and the calculated self-sufficiency degrees for the current

consumption (Table A9) and for a diet based on the PHD (Table A10). The overall self-sufficiency degree (the available ha in relation to the ha necessary for a plant-based diet) regarding current consumption patterns is between 75% (region 2) and 324% (region 4), and it is between 76% (region 2) and 326% (region 4) in terms of PHD recommendations. For single crops, however, the picture looks different. The cultivation of cereals is sufficient in all the regions, and is between 158% (region 2) and 748% (region 4). Sugar beets (considering that 5 kg of sugar beets are necessary to produce 1 kilo of sugar) are cultivated more than is currently needed in region 1 (137%), region 2 (115%), region 4 (264%) and region 5 (130%), but not in region 3 (74%) and region 6 (66%). The PHD recommends eating only one-third of the current sugar intake; thus, for PHD consumption, the amount of cultivated sugar beets would be more than enough in all the regions (region 6: 219%; region 4: 883%).

The SSD of potatoes is between 22% (region 6) and 56% (region 2), which increases to 83% (region 6) and 212% (region 2) based on the PHD. The SSD of oilseeds (considering that 2.3 kg of seeds are necessary to produce 1 kg of oil) is currently between 17% (region 5) and 109% (region 4). These numbers increase to 25% (region 5) and 159% (region 4) when considering PHD intake. The share of legumes not cultivated for livestock can currently only cover between 4% (regions 2 and 5) and 24% (region 4); considering the PHD recommendations, the share would decrease to 0.4% (regions 2 and 5) and to 2.6% (region 4). The SSD of vegetables is between 3.7% (regions 3 and 6) and 43% (region 2), with a slight increase up to 4.3% (regions 3 and 6) and to 50% (region 2), when considering the PHD.

Whereas the land for a locally based and plant-based diet is available, the statistical data shows that the currently cultivated types of arable crops are inadequate to provide the population with a varied and healthy diet (RQ 3). The figures vividly illustrate how the regional farmers' cropping plans do not adapt to the regional needs, but to existing livestock, the global market, subsidies, and arable crops that are less labour-intensive. The insignificant share of the other cereals (about 0.5% of the total cereal production in all the regions), such as summer cereals, millet and sorghum, and non-cereals such as buckwheat and amaranth, all of which are important for a healthy, balanced and varied diet, supports this statement.

### 3.7. Land Consumption Due to Consumption Patterns—Total and per Capita

As stated above and displayed in Figure 5, the pastureland is currently underutilised (region 3: 62%; region 4: 80%). In *SSD 100%*, only in region 3 (89%), region 4 (80%) and region 6 (100%), would there be enough pastureland available to feed the necessary livestock (cf. Table S8). Pastureland would be significantly underutilised in the case of *PHD* (i.e., 14% for region 4 and 65% for region 5).

In other words, each inhabitant needs approx. 767 m$^2$ of pastureland for their current dietary habits (slight deviations per region are due to the different numbers of equines included in the calculation for pastureland), but the available land varies between 185 m$^2$/capita (region 5) and 974 m$^2$/capita. The unequal distribution does not balance out for the whole state of Hesse, and each inhabitant has 467 m$^2$/capita instead of the necessary 767 m$^2$/capita. When considering the PHD recommendations, this share decreases to approx. 128 m$^2$. In this case, no region could live beyond its means.

By combining the share of the cropland necessary for the plant-based diet and the share of the cropland necessary to feed livestock, we can visualise if and to what extent the regions could live beyond their means (Figure 6). Based on the resources necessary to feed current livestock plus the share of current plant-based consumption shares, the cropland would be sufficient only in region 3 (available: 929 m$^2$/capita; necessary: 905 m$^2$/capita; respectively, 97% of cropland necessary in comparison to the existing cropland), region 4 (100%) and region 6 (85%). In the case of *SSD 100%*, only region 4 (available: 1476 m$^2$/capita; necessary: 1212 m$^2$/capita; respectively 82% of the cropland necessary in comparison to the existing cropland) would have enough cropland to directly feed livestock and humans. Region 1 (the whole state of Hesse) exceeds its cropland resources by 81% (in the case of

*SSD 100%).* In *PHD*, most regions, except regions 2 (154%) and 5 (148%), can supply the demand (region 3: 56%; region 4: 37%; region 6: 50% of cropland necessary in comparison to the existing cropland).

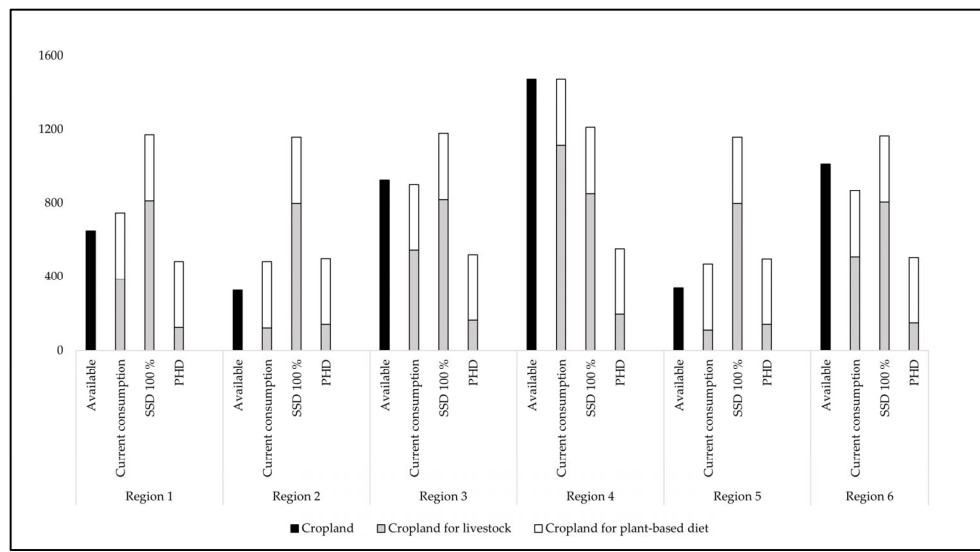

**Figure 6.** Available Cropland in Comparison to Cropland Necessary for Plant-based Diet plus Cropland Necessary to Feed Livestock in m2 per Capita in Regions 1–6. The left bar shows the available cropland; the second bar displays the cropland necessary to cover the share of the plant-based diet and to feed current livestock; the third bar shows the cropland necessary to completely ensure local consumption patterns; and the fourth bar illustrates the cropland necessary to ensure consumption as recommended by the PHD.

Most importantly, if all the Hessian inhabitants consumed as the PHD recommends, the resources would be enough (74% of the cropland necessary in comparison to the existing cropland). Each inhabitant would have 648 m$^2$ of cropland but would only need 482 m$^2$.

Considering that, mathematically, each of us has only 2000 m$^2$ of cropland for our total consumption (including bread, rice, potatoes, fruit, vegetables, oil, sugar, nuts, etc.; drinks, such as juice, beer, wine, etc.; animal products, such as meat, dairy and eggs; cotton, linen, etc., for our clothes; tobacco for smokers; and bio-gas or bio-diesel and renewable raw materials for industrial purposes [88]), the usage of nearly 1200 m$^2$ (*SSD 100%*) of cropland could be considered a non-balanced share. A change in behaviour towards the PHD, or, in general, an adaptation of one's diet consisting of fewer animal products, seems to be imperative if regions do not want to continue to 'live too large'.

At the same time, our data shows that not much would be achieved if only the diet changed, and not farmers' cultivation plans. Regarding the individual crops, therefore, in the case of *PHD*, above all, the pastureland would not be used efficiently, and the local plant-based diet of humans would not become more varied. Rather, more pastureland and crops could theoretically be exported or burned. While the spread of the planetary health diet would make a positive contribution to a more sustainable per capita consumption in ha, a more local diet would be relatively one-sided; moreover, the share of cropland required for the production of meat, dairy and eggs would continue to compete with direct human food (plants), as animals would continue to receive a large share of human-edible cereals and other arable crops. A certain degree of independence from global food chains and a development toward food sovereignty is possible in Hesse, but only if the consumption of animal products is drastically reduced. Is it possible to switch (back) to peasantry farming, and thus agroecological practices, as demanded by La Via Campesina and other organisations, to ensure more environmentally friendly diets, while at the same time agriculture becomes more sustainable and environmentally friendly (cf. RQ 4)?

*3.8. 'What Would Change If . . . ': Self-Sufficiency Degrees Based on the Planetary Health Diet plus Crop Rotation and Extensive Animal Husbandry*

With the collected data and the performed analyses, we can determine the limits of the studied regions (feeding everyone a varied/diverse diet), the impact of the livestock that the regions currently keep, the impact of the livestock that the regions should keep to achieve an SSD of 100% and the resulting impact in the case where everyone eats according to the PHD. Although in the latter case (*PHD*) there would be sufficient land for the reduced number of livestock and animals would continue to receive a significant share of human-edible cereals and other arable crops. Thus, in order to make agriculture more sustainable, animal feeding should also change, even if this means decreasing the production output of livestock (which would not be an issue under a widespread adoption of the PHD).

In Europe alone, the assumed soil loss is around 970 million tonnes per year due to erosion. Because of this and the reasons mentioned above that advocate for the (re-)localising and greening of food chains, a sensible step could be to shift agricultural practices to more sustainable ones. However, what exactly would change, and is there enough land available for (more) extensive livestock farming? To analyse this scenario, we chose a different, utopian-based method. We elaborated a seven-year crop rotation system based on expert input [68] and the literature [89,90]. The crop rotation system consists of two years of clover grass and lucerne followed by one year of high-yielding plants, such as winter wheat, sunflower or rapeseed. Year four is for growing potatoes, oat or medium-to-low-yielding vegetables. The fifth year is for growing grain legumes, such as soya, sweet lupines, chickpeas or lentils, followed by high-to-medium-yielding plants such as sugar beets, sunflower/rapeseed, vegetables, winter wheat and green or silage maize, in year six. The seventh year closes the rotation crop system with a low-yielding cereal, namely, oat, barley or rye. It must be stated that we did not include the different soil types and qualities—which can be more or less suitable for the cultivation of individual arable crops—nor food waste.

Next, we separated the available ha for direct human consumption and the available ha for animal feeding; firstly, we considered the ha necessary for a plant-based self-sufficiency degree of 100% based on the PHD, and secondly, we subtracted the shares remaining after oil and flour processing. The remaining shares were considered for animal consumption (cf. Table S12). We then decreased the output (milk, eggs and animal growth rate; cf. Table A6), because in this utopian scenario, the pastureland and lucernes mainly remain for animal feed. The 'output' of dual-use animals was also included. This reduction would entail an increase in the required number of livestock in contrast to the *PHD* scenario (cf. Tables S10 and S11).

As mentioned above, the necessary $m^2$/capita for the plant-based consumption share based on the PHD and 2150 kcal adds up to 358 $m^2$/capita. The available cropland per capita based on a seven-year crop rotation system would vary between 206 $m^2$ (region 2) and 955 $m^2$ (region 4). Regarding the whole state of Hesse (region 1), each person would have 420 $m^2$ of cropland (instead of the 521 $m^2$ potentially available for direct human consumption without a crop rotation system) for a plant-based diet and 318 $m^2$ of cropland for animal feeding (cf. Table S13, mainly consisting of clover grass and lucernes, 30% wheat that farmers cannot sell due to its quality, corn, production-related sugar and oil residues). To feed the number of livestock necessary based on the PHD, 305 $m^2$/capita of cropland and 393 $m^2$/capita of pastureland (available pastureland: 467 $m^2$/capita) would be necessary in region 1. In percentage, land consumption for the plant-based share and for the consumption of animal products of extensively kept livestock would be 85% and 96% of the available cropland, respectively (total of 81%), and 84% of the available pastureland. Regions 3, 4 and 6 could also satisfy local demand, but the population-intensive areas (i.e., regions 2 and 5) could not meet local demand (both in terms of cropland and pastureland) and would have to be supplied by other Hessian regions. Figure 7 illustrates the available pastureland in comparison with the necessary pastureland (*extensive*) and the available cropland in comparison with the necessary cropland (*extensive*), including its division

between direct human consumption and animal feed. If everyone adapted the PHD and farmers adjusted their crop rotation plans accordingly, our calculations show that despite higher livestock numbers, more extensive livestock production would be possible while still feeding the region adequately.

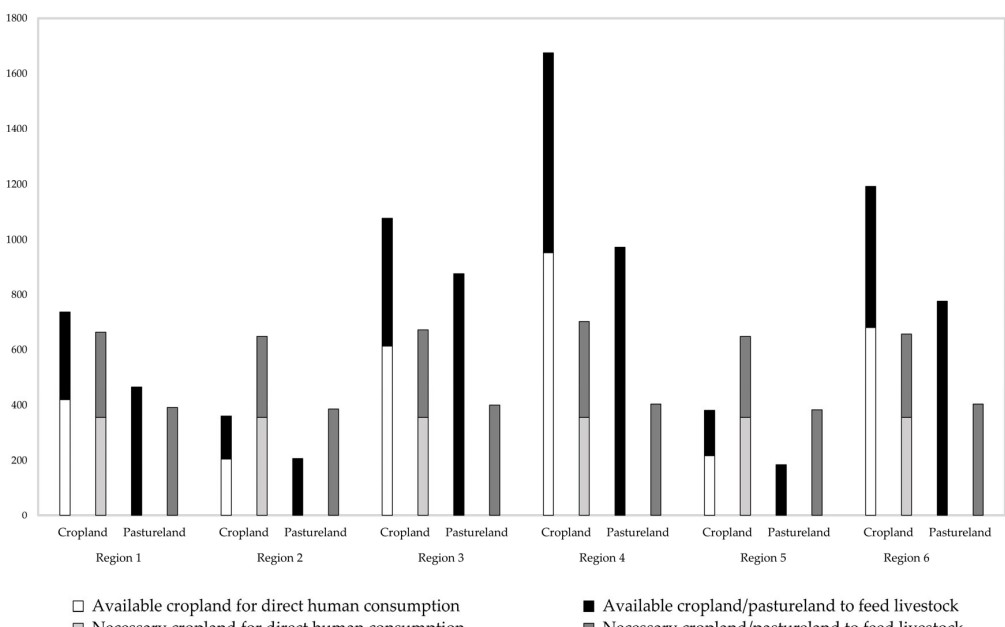

**Figure 7.** Shares of Available Cropland in Comparison with Cropland Necessary for Direct Human Consumption (PHD) and for Animal Feed (Extensive) in Regions 1 to 6 in $m^2$/capita. The shares were calculated based on a seven-year crop rotation system and extensive husbandry (feed only lucerne/glover grass; remaining crops not suitable for human consumption).

Regarding each crop in particular, the main challenge would be to meet the demand for local oil from oilseeds, which would not be satisfied, based on our calculations (cf. Table S13). The proportion of fatty acids in rapeseed is 45%, so only this proportion can be assumed for the supply of oilseed crops. Based on the crop rotation assumed here, the supply of vegetable oils from the region would not be ensured. However, as cropland, in reality, is not divided into 49 shares—as was mathematically performed in our scenario—but is divided by farm and as the farms' crop rotation systems are smaller, the possibility of meeting demand exists—so much that arable land could be used to grow other interesting arable crops to meet the need for seeds (such as hemp seeds, linseed, and pumpkin and sunflower seeds as replacements for the PHD-proposed need for nuts; cf. Table S13: total land consumption related to available vs. necessary cropland and pastureland in region 1: 81%).

Taking into consideration the amount of animal products based on the PHD and the 2150 kcal per day per person, the number of animals could be decreased from the current 0.33 animals/capita to approx. 0.03 animals/capita for red meat and from approx. 0.62 poultry/capita to 0.43 poultry/capita (*SSD 100%* in comparison to *PHD*, including herd factors/stable places). In other words, currently, one dairy cow can satisfy the demand of 17 people, but it could satisfy that of nearly 90 if the overall consumption decreased in line with the recommendations of the PHD. If these animals were kept extensively, and thus the milk yield per cow decreased, the number of cows would increase (respectively, one cow could then satisfy the demand of 49.2 people instead of 90 people). The land for extensive feeding would be available, based on PHD consumption patterns.

The effect on cattle is even greater: whereas, today, nearly 10 people eat one cow per year, one cow could feed 103 people when considering PHD consumption (sheep: 27 vs. 256 people; goats: 27 vs. 258 people; pigs 6 vs. 60 people). In the case of more extensive

livestock production, these numbers would also decrease again to 82 persons per cattle, and 45.3 persons per pig. The numbers for goats and sheep would not change, since even today these animals are mainly kept extensively in the studied regions. Regarding laying hens, the shares would more than triple (1.2 people in *SSD 100%* vs. 4.4 people in *PHD*, though falling to 2.8 persons per hen when kept more extensively). Regarding white meat (broiler chickens and other poultry), the differences are less significant, as the PHD 'allows' rather large amounts of white meat to be consumed per year; whereas, to date, one broiler chicken (occupying one stable place) could meet the demand of 1.8 people, it could meet the demand of 2.6 people (*PHD*) or 1.2 people (*extensive*), and other poultry could satisfy the demand of nearly 22 people, compared with the current demand of 15 people (no difference calculated for *extensive*, partly because the regionally produced quantities are relatively insignificant for current consumption). Figure 8 aims to illustrate these differences.

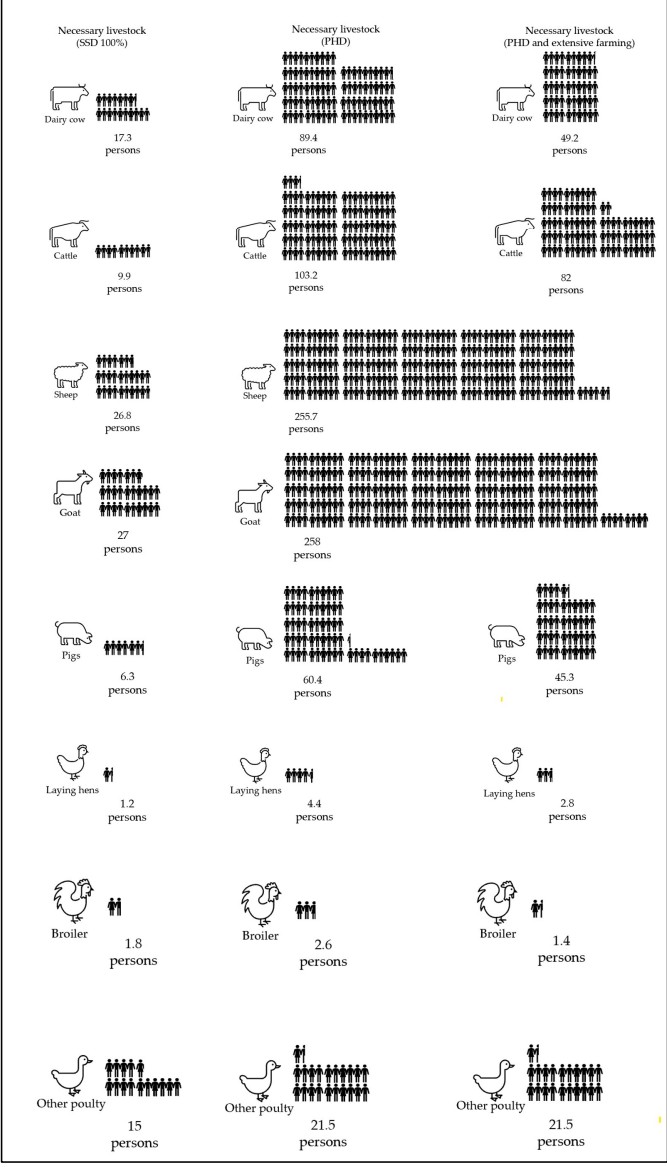

**Figure 8.** Number of Persons one Animal Can Feed (Meat, Dairy, Eggs) in *SSD 100%*, *PHD* and *Extensive*. For instance, currently, one cow ensures the annual dairy consumption of 17.3 persons, but it could ensure the consumption of nearly 90 persons if everyone consumed as recommended by the PHD. If livestock were kept extensively, milk performance would decrease; thus, one cow would only ensure the consumption of about 50 persons, etc. There are no differences in goats, sheep and other poultry between *PHD* and *extensive*.

In brief, our calculations confirmed that a local, diverse and sustainable diet within planetary boundaries and based on peasantry and agroecological farming is possible (RQ 4) if the population drastically reduces its consumption of animal products, farmers cultivate on the basis of crop rotation systems, animals are kept (more) extensively again and dual-purpose animals find their way back into our consumption behaviour.

## 4. Discussion—We Must Change Our Consumption Level

We wanted to visualise the status quo and make the impact of our diet comprehensible. Our results of the calculated self-sufficiency levels based on the current consumption patterns show that regions 1 to 6 are not able to feed the local population and are thus far from food sovereignty or independence from global supply chains. The current prevailing consumption behaviour does not allow these regions to turn away from industrialised agriculture, nor to transform their food system into peasant farming on a broader scale. We could also show that even though consumption patterns changed as recommended by the planetary health diet, the currently cultivated type of arable crops are not sufficient to provide the population with a varied and healthy diet. These are important results because the demand for short food supply chains and the (re-)localisation of food cannot be met with the status quo, nor with changing only one side (here, the consumption behaviour). Ergo, if the status quo prevails, short food supply chains can only be successfully built for small consumer groups, but not for the masses, including out-of-home catering, which, in Germany alone, serves 17 million persons daily [63,65]. Thus, our second goal was to explain if and how food supply and demand can develop together toward sustainability. This was done on the basis of our utopian-based scenario analysis: *extensive*. A diet, such as the PHD (low rates of sugar and animal products), in combination with sustainable farming practices, would not only be good for the environment, biodiversity and soil, but also lead to healthier lifestyles and lower rates of obesity in the population. We emphasise that sustainable farming practices are key for mitigating climate change, as previous studies confirm [13–15,40].

Cultivating using conventional agricultural methods is neither sustainable nor conducive toward gaining independence from big corporations and countries producing fertilisers and pesticides. Research on agroecology already indicates that sustainable farming practices could feed the world, especially when using leguminous cover crops as fertiliser [91], and at the same time, combat climate change [15,92,93]. To arrive there, the status quo methods of producing food need to be reconsidered. In the case of vegetables, for instance, research from the UK indicates that gardening lots and small market gardening farms can produce significant yields per $m^2$, using ecologically sound cultivation methods and aids, such as owning compost, sheep wool pellets, etc. [94]. More research should be performed on compost and biochar as alternatives to animal manure, as the calculated amount of nearly 215,000 grazing animals (scenario *extensive*, region 1) plus pigs is not enough to fertilise 464,000 ha of cropland and nearly 300,000 ha of pastureland (according to experts on organic farming, one livestock unit is needed per hectare for fertilising [68]). To increase yields/$m^2$, among other reasons, it is imperative to undertake further investigations into the following areas: sustainable farming, including traditional practices such as mixed cropping, intercropping systems and no-till farming; the impact of different soil microbiomes on plants; innovations such as precision farming (farming 4.0); artificial intelligence (crop management and risk recognition); genetic engineering (faster maturation, insect and drought resistance); and modern plant breeding (such as Riceberry Rice) [95,96]. Our results support this research. We performed calculations with conventional yields taken from statistical reports. If we had calculated using the current organic yields, we could not mathematically have achieved 100% self-sufficiency on the basis of the planetary health diet, as these are, as of today, about 60% of the conventional yields in the regions analysed [68].

The rigid inclusion of yields per ha could be interpreted as one of the weak points of our model, because land consumption increases or decreases depending on the yields per

ha included, with varying results. Another flaw might be that the studied area is rather small; the variables considered, such as the fodder examples, might significantly change in other areas. In addition, the proposed crops are certainly not suitable for all areas because of the different regional cultivation conditions. Furthermore, our analyses did not take into consideration the issue of food waste, nor did it account for the higher cost of local products in comparison to imported ones, which is often the determining factor behind consumer purchasing decisions, rather than mere product availability. Moreover, our study only considers land consumption. Other resource consumption of agriculture or other relevant environmental impacts, such as water use [97], was not considered in our study. Other relevant environmental impacts should play an important role in future studies.

Having stated in the first section that people love and act upon metrics, such as carbon footprints, much more than upon doomsday messages or abstract models not in touch with local circumstances, we believe that we provide important figures in this study. We can clearly demonstrate how much each of us consumes and what would change if we ate differently. Illustrative results—such as the land consumption in square meters per capita—are important in activating behavioural change processes in people. By including local data for local people, results become more tangible than results globally produced using abstract models. This, together with the combination of production and consumption, we believe, is the strength and the novelty of our study. Our framework demonstrates an approach for other regions to determine their level of self-sufficiency using regional data. Though created for and adapted to the region of Hesse, it can easily be adapted to other regions, mainly by adapting fodder examples, agricultural statistics and population size. Further, it can be set in broader contexts, such as Germany or Europe, because the consumption patterns and diet behaviour are similar, if not the same.

*4.1. Climate Change Mitigation—A Multi-Layered Global Challenge*

For a couple of decades, demand and, thus, supply have developed in an economic but unsustainable direction, making food systems now responsible for about one-third of total greenhouse-gas emissions. With Greta Thunberg's words in mind: '[ . . . ] I want you to panic. I want you to act as if your house was on fire [ . . . ]' [98], action has to be taken in many areas at the same time. To quickly reduce greenhouse-gas emissions, and at the same time, achieve the two-fold objective of a healthier and well-nourished humanity, we urgently need to transform our food systems. One important matter is to quickly reduce the consumption of animal products [22,23]. Global challenges on this issue are multi-layered and need to be approached from different angles simultaneously.

One important step to mitigate climate change could be the redirecting of the Green Revolution toward peasantry and agroecology farming, prioritizing food for the people and not neo-liberal policies and international trade. That this could be a potentially viable path is indicated by our results in the *extensive* scenario. To achieve this, we propose getting farmers back into thinking in terms of a circular economy and cultivating food for people rather than for livestock. As farmers adapt their farming methods to the political will and are demand-driven, both policies and demand must change. Policymakers have great leverage, as they can implement and enforce new laws and policies and conduct wide-reaching campaigns for change. In some political parties, this process has been started. The German government has published a nutrition strategy focusing on a climate friendly and healthy diet for all. This is to be achieved through a 'systemic approach of behavioural and prevention, which takes into account the effects on the environment and climate and the different lifestyles [ . . . ]' [14]. The EU commission has published its Common Agricultural Policy (CAP) 2023–27, seeking to ensure a sustainable future for European farmers and providing more targeted support to smaller farms while focusing on climate change action, environmental care, preserving landscapes and biodiversity etc. [99].

Supporting and rewarding farmers is important because farmers, as a homogenised group, are not known for perceiving climate change as man-made and acting accordingly,

with 66% of 4778 farmers surveyed believing that climate change is happening and only 8% attributing it to human activity [100].

However, to persuade politicians to make new laws and policies, they need to be equipped with a global and local scientific background. Calculations based on local data, as presented in this study, are helpful to directly show policymakers what economic and environmental impacts their policies may have on a region. Policymakers taking care of daycare centres and schools could have a huge impact, as in German daycare centres and schools, each year, about 1.2 billion lunches are offered [65,101]. If it became mandatory to cook as recommended by the PHD, children would be fed not only healthily but also in an environmentally friendly manner. If canteens supplied the estimated 700 tonnes of ingredients locally and demanded sustainable production, farmers signalled in discussions that they would adjust their cropping plans to meet this demand, as long as they were compensated fairly for doing so. The side effects of this transition might be to get young people excited again about (good) food production, bringing back local processing businesses such as mills, oil presses, slaughterhouses and butchers, and to make regional logistics efficient.

In general, out-of-home catering can help discover new ways to eat; therefore, canteens and catering businesses must change their menus. The healthy, sustainable menus they have must provide a sense of well-being and enjoyment, but still be filling. Guests should be encouraged to try new things. To decrease the consumption of animal products and thus, land consumption, people need to be made aware of the matter on multiple levels, such as through nudging, campaigning, experiencing and understanding the urgency to do so.

Whether this change is really going to happen or will happen fast enough is highly questionable. Not only are humans creatures of habit, also not all policies are working in the direction of (re-)localising food. For instance, the European Union works on free trade agreements, such as Mercosur, which ensure that more and not fewer agricultural products, including cattle and resources for feeding livestock, are transported from long distances and at low prices [102]. If the prices of products from small regional farmers remain many times more expensive than industrial, tax-privileged products from third countries, how is a change in mindset of price-sensitive consumers supposed to take place? What is the answer of politicians to their own counter-productive policies to such urgent questions as revolutionising our global food system?

Before concluding the discussion, we would like to address two issues that are somewhat disconnected from the main focus of this study yet are important: the adaptation of the planetary health diet to local conditions and the impact of pets on the environment.

### 4.2. Adapting the Planetary Health Diet to Local Cuisine and Culture

We were able to indicate how land consumption would decrease if everyone's consumption followed the planetary health diet recommendations, at least for a certain range of food groups. We are aware that we excluded a wide range of foodstuff, such as fruits, nuts and fish; beverages, such as coffee, tea, juice, wine, beer, etc.; processed food and sweets, such as ice-cream, chocolate and cookies; snack food, such as crisps, etc.; and tobacco and other luxury foods. The implication of these exclusions is two-fold: firstly, the resource consumption necessary for our entire diet alone is much larger than that shown in this study; secondly, if it is that much larger, should each of us not try to keep the footprint of staple food as small as possible, so that occasionally, we can enjoy other foodstuffs without constantly stretching planetary boundaries? The planetary health diet can give us good direction, as it indicates a healthy diet that is within our planetary boundaries. Nonetheless, it seems important that we adjust it to our local cuisine and culture, and cooking time and capabilities, especially if breaking consumption patterns is the goal—a project that will meet much resistance from the population, as people do not like to feel patronised.

In the case of the state of Hesse—or rather central Europe as such—the medium potato consumption indicated by the PHD is too low. Instead, we propose to maintain the current consumption levels, as in this region, potatoes are usually eaten whole and only to a lesser

extent in the form of starch (the reason why the PHD does not recommend greater potato consumption). To maintain the recommended calorie intake, the consumption of cereals, for example, could be reduced accordingly. The recommended consumption of poultry at 3.6 animals per capita per year is quite high. This number of poultry can only be produced through factory farming, a form of husbandry that, as critics argue (and we could not agree more), cannot do justice to animal welfare. It might therefore make more sense to use this land to grow special crops such as hemp, flax, buckwheat, quinoa, chickpeas, etc., instead of feeding them to poultry, to provide a more varied plant-based diet at the local level. The same also applies to fish, although it is not part of this study due to the insignificant amount of locally produced fish: is the annual intake of almost 9 kg of fish per capita necessary, or could we not rather consume similar healthy food, such as local plants, but also algae and equivalents? Examples of balanced, varied vegan diets prove that sufficient consumption of proteins deriving from plants is possible [103].

In most parts of the state of Hesse, the current production of fruits and especially nuts is irrelevant, besides perhaps cherries, strawberries and apples, where the latter are employed to produce beverages. By researching local gardens, however, we can find that tonnes of fruits are produced which often rot on trees and bushes. This practice could be reconsidered by making the advantages of local fruit varieties in season or preserved fruit socially attractive again. The PHD also recommends a high intake of nuts (15.7 kg per capita per year). Most nuts—such as almonds, cashews, Brazil nuts and peanuts—are not native to Germany, and their cultivation requires a large amount of water. Instead of campaigning to eat more nuts, the focus could be on local nuts, such as hazelnuts and walnuts, but especially on the consumption of local seeds, such as hemp seeds, linseed, pumpkin seeds and sunflower seeds.

*4.3. The Planetary Health Diet and Pets*

Another important aspect, which should not be neglected and which could hinder the achievement of the sustainability goals in the food sector—are pets (cats and dogs). In Germany, there are 16.7 million pet cats and 10.3 million pet dogs, and the share for the population of Hesse is about 770,000 dogs and 1.25 million cats. For cats (mean weight of 3–5 kg), this means roughly 55,000 tonnes of meat per year, and for dogs, based on the weight of the ten most popular dog breeds and the mean value of different feeding practices (besides so-called raw feeding), about 13,000 tonnes of meat, 3100 tonnes of vegetables and 3100 tonnes of cereals of feeding are demanded. Converted into land used to feed the animals being fed to pets, and considering cattle, poultry and sheep (including the herd factors/stable places reported in Table A6), an average dog needs nearly 2000 m$^2$ of cropland and 2000 m$^2$ of pastureland per year. Cats need more meat and use about 6000 m$^2$ of cropland and about 7000 m$^2$ of pastureland, which is, as such, a multiple of the land consumption of human beings (especially adopting the PHD). To date, these numbers can be relativised, because most of the meat consumed by pets is slaughter waste (including bones) or animals, not (or less) eaten by humans, such as brother cocks or retired dairy cows. Thus, the amount of additionally raised livestock for pets is currently much smaller. What would happen, however, if all humans ate as the PHD recommends? Then, far fewer animals would have to be slaughtered, and, consequently, less slaughter waste would be produced. How could the number of pets then be fed? Provocatively speaking, should we not, as a society, outgrow the idea of keeping animals for our pleasure, when millions of people go hungry and we need three worlds to sustain our current consumption patterns?

**5. Conclusions**

Organic farming practices, advocates of the status quo argue, is not an option, because yields are too low. When Russia invaded Ukraine, there was a huge discussion that the prices of grain and sunflower oil would skyrocket [104], while, simultaneously, what is seen on German fields is wheat and rapeseed. Catering businesses claim that local produce is not available, so they have to buy it on international markets.

We were curious about whether these discussions and claims were really true. We started looking at statistics but could not make any sense of them. What do the indicated hectares mean for the local self-sufficiency levels? What are the differences between the North and the South of the state of Hesse? Given this initial premise, we performed our calculations. First, we calculated different self-sufficiency degrees for animal products (meat, eggs, dairy) and plants (cereals, legumes, vegetables, oil, sugar, potatoes) and were proven right: self-sufficiency levels are way too low for current consumption behaviour. Then, we calculated the number of livestock necessary to reach a self-sufficiency level of 100% for animal products, and compared the land consumed for feeding this livestock (*SSD 100%*) with the land used to feed the current livestock. In both scenarios, the land necessary for feeding exceeded the available cropland and pastureland. Thus, we adapted this method to the scenario where everyone ate as the planetary health diet recommends and could substantially decrease the number of livestock. The results show that although land to feed animals (*PHD* livestock) and for direct human consumption is available, we could observe that farmers' practices are neither sustainable nor able to feed us properly. Based on these results, we developed the utopian-based model (a seven-year crop rotation and extensive husbandry) just to see what could be possible if everybody acted differently. With our calculation results, we were able to demonstrate the potential for positive change: mathematically, in this scenario, most of the regions are able to feed themselves a balanced plant-based diet and keep the necessary number of livestock in an extensive and sustainable way. However, more livestock is needed because of the consequently lower production rates.

Though not in scope of this study, the question remains about how things could change. This question is one of the most important of our time. To 'extinguish the burning house', to rephrase Greta Thunberg's words, we believe that we need an army of tools for this challenge. As stated above, food production accounts for about one-third of total greenhouse-gas emissions, so acting in this sector is key. Our contribution to this massive project is our model, which can be easily adapted to user-friendly tools, such as an app that shows one's personal land consumption based on consumption behaviour, including smiley faces and frowning faces (as we have indicated above that people love them). Our figures and data can be used for campaigns and round-table discussions; policy makers can use them to convince their opponents to start changing our food systems. Even though the challenges are immense and may appear impossible to overcome, it is crucial that we persist in our efforts.

**Supplementary Materials:** The following supporting information can be downloaded at: https://www.mdpi.com/article/10.3390/su15118675/s1, in form of an Excel file and include Table S1: Feeding examples; Table S2: Livestock feed; Table S3: Crops; Table S4: Calc. base energy plants; Table S5: Livestock; Table S6: SSD plants; Table S7: Clover grass % Hesse; Table S8: SSD animal products; Table S9: SSD 100% Livest feed; Table S10: PHD Livestock feed; Table S11: PHD extensive Livest feed; Table S12: Overview consump, production; Table S13: PHD incl crop rotation. References [105–109] are mentioned in Supplementary Materials File.

**Author Contributions:** Conceptualization, A.-M.S. and M.B.; methodology, A.-M.S.; validation, A.-M.S. and M.B.; formal analysis, A.-M.S.; investigation, A.-M.S. and M.B.; resources, A.-M.S. and M.B.; data curation, A.-M.S. and M.B.; writing—original draft preparation, A.-M.S.; writing—review and editing, A.-M.S. and M.B.; visualization, M.B.; supervision, A.-M.S.; project administration, A.-M.S. and M.B.; funding acquisition, A.-M.S. and M.B. All authors have read and agreed to the published version of the manuscript.

**Funding:** This research was partly funded by Innovations in the field of logistics and mobility measure of the Hessian Ministry of Economics, Energy, Transport and Housing, grant number 1268/21-169.

**Institutional Review Board Statement:** Not applicable.

**Informed Consent Statement:** Not applicable.

**Data Availability Statement:** Publicly available datasets were analysed in this study. This data can be found here: https://statistik.hessen.de/unsere-zahlen/land-und-forstwirtschaft (accessed on 12 September 2022) and here: https://de.statista.com (accessed on 14 December 2022).

**Acknowledgments:** We would like to thank Christoph Feist for sharing his knowledge on agricultural practices. We would also like to thank Kenan A.J. Bozhüyük and Susanne von Münchhausen for reading, commenting on and discussing our work.

**Conflicts of Interest:** The authors declare no conflict of interest.

## Appendix A

**Table A1.** Available Crop Land per Regions 1 to 6 in ha and Yields per Region and Crop in decitonnes per hectare. Note: One decitonne (dt) is equivalent to 100 kg or one quintal. Each region (1–6) is divided into the total hectares of each arable crop and the corresponding organic shares in hectares and percent. Table reprinted/adapted with permission from Ref. [54]. Copyright year: 2023, copyright owner: Hessisches Statistisches Landesamt.

| Available Crop Land in Hectare, Regions 1 to 6 (I) | | | | | | | | | | | |
|---|---|---|---|---|---|---|---|---|---|---|---|
| | | | | | | thereof | | | | | |
| Region | | Operating Farms | Crop Land Total | Cereals Total | Wheat, Spelt, Einkorn | Rye | Triticale | Barley | Oat | Corn Maize/Corn-Cob-Mix | Other Cereals |
| 1 Hesse | in total | 15,128 | 464,437 | 289,347 | 143,606 | 15,059 | 19,342 | 87,266 | 9277 | 13,470 | 1327 |
| | thereof organic | 2108 | | 22,296 | 9318 | 2383 | 3101 | 3655 | 2496 | 787 | 556 |
| | in % | 14% | | 8% | 6% | 16% | 16% | 4% | 27% | 6% | 42% |
| 2 GD Da | in total | 4935 | 145,714 | 88,511 | 48,442 | 4670 | 2440 | 23,057 | 2458 | 6963 | 481 |
| | thereof organic | 486 | | 4506 | 1925 | 467 | 372 | 712 | 534 | 389 | 108 |
| | in % | 10% | | 5% | 4% | 10% | 15% | 3% | 22% | 6% | 22% |
| 3 GD Gi | in total | 3832 | 113,216 | 72,640 | 33,400 | 3658 | 5657 | 23,318 | 3039 | 3198 | 370 |
| | thereof organic | 743 | | 8677 | 3523 | 962 | 1292 | 1402 | 1060 | 235 | 202 |
| | in % | 19% | | 12% | 11% | 26% | 23% | 6% | 35% | 7% | 55% |
| 4 GD Ka | in total | 6361 | 204,239 | 128,197 | 61,764 | 6731 | 11,244 | 40,892 | 3780 | 3310 | 476 |
| | thereof organic | 879 | | 9113 | 3870 | 954 | 1437 | 1541 | 902 | 163 | 247 |
| | in % | 14% | | 7% | 6% | 14% | 13% | 4% | 24% | 5% | 52% |
| 5 F MPA | in total | 3207 | 112,825 | 69,097 | 38,528 | 3991 | 1825 | 16,787 | 1614 | 5967 | 385 |
| | thereof organic | 323 | | 3691 | 1578 | 392 | 282 | 480 | 378 | 330 | 65 |
| | in % | 10% | | 5% | 4% | 10% | 15% | 3% | 23% | 6% | 17% |
| 6 M-B | in total | 1106 | 29,664 | 19,765 | 8189 | 1417 | 1791 | 5777 | 889 | 1619 | 83 |
| | thereof organic | 197 | | 2625 | 918 | 370 | 449 | 471 | 277 | 103 | 35 |
| | in % | 18% | | 13% | 11% | 26% | 25% | 8% | 31% | 6% | 42% |

| Available Crop Land in Hectare, Regions 1 to 6 (II) | | | | | | | | | |
|---|---|---|---|---|---|---|---|---|---|
| Region | | Silage Maize | Sugar Beets | Potatoes | Winter Oilseed Rape | Pulses | Vege-tables | Grass-land | Per-manent Crops | Clover Grass/Lucerne |
| 1 Hesse | in total | 43,897 | 16,504 | 4421 | 43,204 | 13,410 | 7494 | 294,288 | 5855 | 501 |
| | thereof organic | 1425 | 318 | 495 | 179 | 4666 | | | | |
| | in % | 3% | 2% | 11% | 0% | 35% | | | | |
| 2 GD Da | in total | 11,689 | 8845 | 3192 | 11,406 | 3303 | 6582 | 83,574 | 4888 | 133 |
| | thereof organic | 188 | 157 | 187 | 52 | 881 | | | | |
| | in % | 2% | 2% | 6% | 0% | 27% | | | | |
| 3 GD Gi | in total | 10,198 | 1486 | 487 | 11,465 | 3737 | 150 | 92,089 | 202 | 116 |
| | thereof organic | 550 | 52 | 103 | 71 | 1595 | | | | |
| | in % | 5% | 3% | 21% | 1% | 43% | | | | |
| 4 GD Ka | in total | 22,010 | 6173 | 742 | 20,333 | 6371 | 763 | 118,627 | 726 | 251 |
| | thereof organic | 688 | 109 | 205 | 57 | 2190 | | | | |
| | in % | 3% | 2% | 28% | 0% | 34% | | | | |
| 5 F MPA | in total | 9155 | 7361 | 1014 | 7773 | 2463 | 4569 | 54,763 | 1400 | 104 |
| | thereof organic | 115 | 69 | 127 | 0 | 762 | | | | |
| | in % | 1% | 1% | 13% | 0% | 31% | | | | |
| 6 M-B | in total | 2490 | 313 | 77 | 2200 | 941 | 35 | 19,311 | 21 | 339 |
| | thereof organic | 152 | 22 | 22 | 0 | 538 | | | | |
| | in % | 6% | 0% | 29% | 0% | 57% | | | | |

**Table A2.** Yields per Crop in dt/ha based on [66].

| Field Crop | Yield dt/ha Hesse (Year: 2021) |
|---|---|
| Cereals total incl. corn maize and corn-cob-mix | 67.9 |
| Cereals total excluding corn maize and corn-cob-mix | 66.7 |
| Wheat (mean value winter and summer wheat) | 65.95 |
| Rye | 56.3 |
| Barley (mean value winter and summer barley) | 64.5 |
| Oilseed rape (winter) | 35.5 |
| Potatoes | 420.6 |
| Sugar beets | 847.3 |
| Corn maize | 93.3 |
| Silage maize | 547.9 |
| Forage (permanent grassland) | 60 |
| Forage (cultivation on arable land) | 61.8 |
| Clover grass/alfalfa (dry mass) | 60.3 |
| Field beans | 37.9 |
| Field peas | 35.4 |
| Sweet lupines | 33.5 |
| Soybeans | 34 |
| Sunflower seeds | 26.1 |

**Table A3.** Yields in dt per Region and Crop based on [54,66]. Table adapted with permission from Ref. [54]. Copyright year: 2023, copyright owner: Hessisches Statistisches Landesamt.

| | | | | | | | | |
|---|---|---|---|---|---|---|---|---|
| **Yields per Region and Crop in dt per Hectare (I)** | | | | | | | | |
| | | **there of** | | | | | | |
| **Region** | **Cereals Total** | **Wheat, Spelt, Einkorn** | **Rye** | **Triticale** | **Barley** | **Oat** | **Corn Maize/Corn-Cob-Mix** | **Other Cereals** |
| 1 Hesse | 19,100,767 | 9,274,372 | 847,822 | 1,275,605 | 5,628,657 | 447,267 | 1,256,751 | 33,175 |
| 2 GD Da | 5,836,665 | 3,128,484 | 262,921 | 160,918 | 1,487,177 | 118,506 | 649,648 | 12,025 |
| 3 GD Gi | 4,798,879 | 2,157,041 | 205,945 | 373,079 | 1,504,011 | 146,518 | 298,373 | 9250 |
| 4 GD Ka | 8,465,290 | 3,988,847 | 378,955 | 741,542 | 2,637,534 | 182,243 | 308,823 | 11,900 |
| 5 F MPA | 4,555,467 | 2,488,218 | 224,693 | 120,359 | 1,082,762 | 77,815 | 556,721 | 9625 |
| 6 M-B | 1,306,996 | 528,863 | 79,777 | 118,116 | 372,617 | 42,861 | 151,053 | 2075 |
| **Yields per Region and Crop in dt per Hectare (II)** | | | | | | | | |
| **Region** | **Silage maize** | **Sugar beets** | **Potatoes** | **Winter oilseed rape** | **Pulses** | **Vegetables** | | |
| 1 Hesse | 21,884,168 | 13,726,924 | 1,852,673 | 1,422,695 | 350,516 | 2,148,383 | | |
| 2 GD Da | 5,827,370 | 7,356,680 | 1,337,646 | 375,596 | 86,335 | 1,886,931 | | |
| 3 GD Gi | 5,084,055 | 1,235,955 | 204,083 | 377,539 | 97,679 | 43,002 | | |
| 4 GD Ka | 10,972,744 | 5,134,289 | 310,944 | 669,560 | 166,528 | 218,737 | | |
| 5 F MPA | 4,564,083 | 6,122,388 | 424,929 | 255,963 | 64,379 | 1,309,843 | | |
| 6 M-B | 1,241,351 | 260,332 | 32,268 | 72,445 | 24,596 | 10,034 | | |

**Table A4.** Extrapolated Consumption per Regions 1 to 6 in kg according to Current Diet and Planetary Health Diet.

| Food Group (in kg) | Current Consumption Region 1 | Consumption Recommended by PHD Region 1 | Current Consumption Region 2 | Consumption Recommended by PHD Region 2 | Current Consumption Region 3 | Consumption Recommended by PHD Region 3 | Current Consumption Region 4 | Consumption Recommended by PHD Region 4 | Current Consumption Region 5 | Consumption Recommended by PHD Region 5 | Current Consumption Region 6 | Consumption Recommended by PHD Region 6 |
|---|---|---|---|---|---|---|---|---|---|---|---|---|
| Cereals | 523,745,414 | 458,433,354 | 335,010,125 | 293,233,718 | 87,423,648 | 76,521,751 | 101,311,642 | 88,677,885 | 245,660,896 | 215,026,510 | 20,681,606 | 18,102,570 |
| Pulses | 15,737,543 | 148,200,438 | 10,066,410 | 94,795,383 | 2,626,913 | 24,737,635 | 3,044,220 | 28,667,420 | 7,381,638 | 69,512,880 | 621,443 | 5,852,124 |
| Potatoes | 375,183,013 | 98,800,292 | 239,983,214 | 63,196,922 | 62,625,594 | 16,491,757 | 72,574,205 | 19,111,613 | 175,978,238 | 46,341,920 | 14,815,189 | 3,901,416 |
| Vegetables | 688,674,860 | 592,801,751 | 440,506,102 | 379,181,532 | 114,953,691 | 98,950,540 | 133,215,067 | 114,669,679 | 323,020,457 | 278,051,521 | 27,194,324 | 23,408,496 |
| Fruits | 453,870,726 | 395,201,167 | 290,315,264 | 252,787,688 | 75,760,157 | 65,967,027 | 87,795,305 | 76,446,453 | 212,886,426 | 185,367,681 | 17,922,402 | 15,605,664 |
| Plant-Oil | 148,562,401 | 102,357,102 | 95,026,910 | 65,472,011 | 24,798,054 | 17,085,460 | 28,737,437 | 19,799,631 | 69,682,658 | 48,010,229 | 5,866,417 | 4,041,867 |
| Nuts | 31,475,085 | 98,800,292 | 20,132,820 | 63,196,922 | 5,253,825 | 16,491,757 | 6,088,440 | 19,111,613 | 14,763,275 | 46,341,920 | 1,242,885 | 3,901,416 |
| Sugar | 204,588,053 | 61,256,181 | 130,863,330 | 39,182,092 | 34,149,863 | 10,224,889 | 39,574,860 | 11,849,200 | 95,961,288 | 28,731,991 | 8,078,753 | 2,418,878 |
| Milk, equiv. diary | 2,553,258,895 | 494,001,459 | 1,633,174,358 | 315,984,610 | 426,190,284 | 82,458,783 | 493,894,253 | 95,558,066 | 1,197,596,868 | 231,709,601 | 100,822,831 | 19,507,080 |
| Eggs (pcs.) | 1,498,214,046 | 407,747,236 | 958,322,232 | 260,812,694 | 250,082,070 | 68,061,218 | 289,809,744 | 78,873,324 | 702,731,890 | 191,252,369 | 59,161,326 | 16,101,082 |
| Red meat total | 264,390,714 | 27,664,082 | 169,115,688 | 17,695,138 | 44,132,130 | 4,617,692 | 51,142,896 | 5,351,252 | 124,011,510 | 12,975,738 | 10,440,234 | 1,092,396 |
| White meat | 82,464,723 | 57,304,169 | 52,747,988 | 36,654,215 | 13,765,022 | 9,565,219 | 15,951,713 | 11,084,736 | 38,679,781 | 26,878,314 | 3,256,359 | 2,262,821 |
| Fish | 79,946,716 | 55,219,889 | 51,137,363 | 35,321,019 | 13,344,716 | 9,217,311 | 15,464,638 | 10,681,559 | 37,498,719 | 25,900,690 | 3,156,928 | 2,180,517 |
| Beef | 514,302,889 | | 328,970,279 | | 85,847,501 | | 99,485,110 | | 241,231,914 | | 20,308,741 | |
| Sheep and goat | 59,173,160 | | 37,849,702 | | 9,877,191 | | 11,446,267 | | 27,754,957 | | 2,336,624 | |
| Pig | 3,777,010 | | 2,415,938 | | 630,459 | | 730,613 | | 1,771,593 | | 149,146 | |
| Other meat | 195,145,527 | | 124,823,484 | | 32,573,715 | | 37,748,328 | | 91,532,305 | | 7,705,887 | |
| Meat total (red and white + industry) | 6,295,017 | | 4,026,564 | | 1,050,765 | | 1,217,688 | | 2,952,655 | | 248,577 | |

**Table A5.** Extrapolated Consumption per Regions 1 to 6 in ha according to Current Diet and Planetary Health Diet.

| Region | Cereals Total (Wheat, Spelt, Einkorn, Rye, Barley, Oat, Other Cereals) | Sugar from Sugar Beets 20% Sugar per Beet | Potatoes | Oil from Oilseed Rape 2.3 kg per 1 L Oil | Legumes (95% in Hesse for Livestock) | Vegetables | Sum |
|---|---|---|---|---|---|---|---|
| **Necessary ha of Crops for Direct Human Consumption for Self-Sufficiency Degree of 100% (Baseline: Current Consumption)** | | | | | | | |
| 1-Hesse | 78,523 | 12,073 | 8920 | 96,252 | 6733 | 24,022 | 226,522 |
| 2-GD Da | 50,226 | 7722 | 5706 | 61,567 | 4306 | 15,366 | 144,893 |
| 3-GD Gi | 13,107 | 2015 | 1489 | 16,066 | 1124 | 4010 | 37,811 |
| 4-GD Ka | 15,189 | 2335 | 1725 | 18,619 | 1302 | 4647 | 43,818 |
| 5-F MPA | 36,831 | 5663 | 4184 | 45,147 | 3158 | 11,268 | 106,250 |
| 6-M-B | 3101 | 477 | 352 | 3801 | 266 | 949 | 8945 |
| **Necessary ha of Crops for Direct Human Consumption for Self-Sufficiency Degree of 100% (baseline: Planetary Health Diet)** | | | | | | | |
| 1-Hesse | 68,731 | 3615 | 2349 | 66,316 | 63,401 | 20,678 | 225,090 |
| 2-GD Da | 43,963 | 2312 | 1503 | 42,418 | 40,554 | 13,227 | 143,977 |
| 3-GD Gi | 11,473 | 603 | 392 | 11,069 | 10,583 | 3452 | 37,572 |
| 4-GD Ka | 13,295 | 699 | 454 | 12,828 | 12,264 | 4000 | 43,541 |
| 5-F MPA | 32,238 | 1696 | 1102 | 31,105 | 29,738 | 9699 | 105,578 |
| 6-M-B | 2714 | 143 | 93 | 2619 | 2504 | 817 | 8888 |

**Table A6.** Herd Factors/Stable Places, Slaughter Weight and Output of Conventional and Extensive Husbandry. Table adapted and based on [67–76].

| Livestock | Herd Factors | Herd Factors (Extensive Husbandry) | Herd Share Producing "Output"/ Stable Place (Slaughter Quota) | Output kg Milk/Animal; Eggs/Animal | Output Slaughter Weight/ Animal in kg | Output (Extensive Husbandry) |
|---|---|---|---|---|---|---|
| Dairy cow | 1.33 | | 0.67 | 9358.4 | | 5150 kg/milk p.a. |
| Cattle | 2.7 | 3 | 0.37 | | 230 | |
| Sheep | | | 0.5 | | 21.4 | |
| Goats | | | 0.5 | | 10.8 | |
| Fattening pigs | | | 2 | | 98 | 1.5 |
| Laying hen | | | | 288 | | 180 eggs/hen p.a. |
| Broiler chicken | | | 10 | | 2 | |
| Geese | | | 4.1 | | 5.2 | |
| Turkeys | | | 2.9 | | 10 | |
| Ducks | | | 4.3 | | 2.2 | |

Additional Information on A6: Herd Factor/Stable Place Factor:

Dairy cows:

The herd factor of dairy cows is 1.33 (dairy cow herd plus 33% offspring), since the offspring give birth to their first calf at about 2–2.5 years of age, thus producing milk, and a dairy cow is slaughtered after approx. 6 years. Before then, the cow has to live in the herd without producing milk. Thus, the share of the herd producing milk is 67%.

We considered the current dairy cows, added 1/3 of offspring and calculated that 67% of this herd produces 9358.4 kg currently or 5150 kg milk/cow p.a. with extensive husbandry [72].

Cattle:

Taking into account Hessian statistics from *Agricultural holdings with cattle husbandry and cattle population on 1 March 2020 by regional unit, Statistics Hesse* [69] and, calculating the decrease in animals in each age group, we calculated a slaughter quota of 0.37. The average slaughter weight of calves, young cattle and cattle is 230 kg/animal. To slaughter 37% of a herd, each slaughtered animal has to be multiplied by 2.7 (1/0.37) to keep the

herd size stable. If cattle were kept extensively, the multiplying factor would have to be 3 (one animal lives for three years to achieve an adequate slaughter quota). To calculate the produced cattle meat in kg based on the current livestock plus the retired dairy cows, we took the current livestock minus the dairy cows, multiplied them by the slaughter quota (0.37) and slaughter weight (230 kg), and 17% of the dairy cows and a slaughter weight of 250 kg.

Sheep and goats:

These differences are less significant for sheep and goats as most farms in the state of Hesse keep sheep on pasture and grass silage for meat production. The number of goats was included, and there are a few dairy goat farms, but the number is rather insignificant, accounting for only 2% of the available grazing animals.

Calculating the number of goats, sheep and other animals is easier than for cattle as they are slaughtered a few weeks/months after being born. We did not consider different livestock for extensive husbandry as most goats and sheep are kept extensively. In addition, the number of animals necessary to satisfy the demand is not high, and as a much lower number of cattle is necessary, grazing land would be available. No broiler chickens would be necessary, as the demand could be covered by dual-purpose hens and their brothers. We included a slaughter quota of 0.5 for sheep and goats, and slaughter weight of 21.4 kg for sheep and 10.8 kg for goats.

Fattening pigs:

The included slaughter weight of fattening pigs is 98 kg. The slaughter quota is 2, as each pig lives for max. 6 months; thus, per stable place, two pigs can be fattened per year. In the case of extensive husbandry, we assumed pigs to be slaughtered after 8 months; thus, the slaughter quota would change to 1.5.

Breeding sows and equines:

The current number of breeding sows per region was included but was not changed in the other scenarios (*SSD 100%*, *PHD*, *extensive*).

Laying hens and pullets:

The current laying performance is, on average, 288 eggs/hen p.a., and for dual-use chickens, 180 eggs/hen p.a. [73]. Pullets were included in the current livestock.

Broiler chickens:

To calculate the currently produced chicken meat in kg, we used the current livestock multiplied by 2 kg slaughter weight (i.e., the average of light, medium, and heavy fattening according to [71]) and 10 (as, on average, broiler chickens are replaced after 37 days). For extensive husbandry, chickens are slaughtered after 81 days (slaughter quota: 4.5).

Other poultry:

Geese, turkeys and ducks account for a 12% consumption share in the region. The slaughter quota of geese is estimated to be 6.1 (slaughtered after 90 days), with a slaughter weight of 5.2 kg; that of turkeys is 3.3 (slaughtered after 126 days), with a slaughter weight of 10 kg; and that of ducks is 2.6 (slaughtered after 84 days), with a slaughter weight of 2.2 kg.

All livestock

All calculations of the necessary livestock (*SSD 100%*, *PHD* and *extensive*) were performed by dividing the production necessary for *SSD 100%* in kg by the output and multiplying by the herd factors (Table A6).

**Table A7.** Calculated Shares of Plants Used for Energy Production. Because this share of arable land is not used to produce food or fodder, it was excluded from calculations from the outset. Table based on [66,82,83].

| Percentages of Crops for Energy Purposes and Industry (Excl. Starch Production) | | | | | | |
|---|---|---|---|---|---|---|
| **Cereals Total** | **Wheat, Spelt, Einkorn** | **Silage Maize** | **Sugar Beets** | **Potatoes** | **Winter Oilseed Rape** | **Pulses** |
| | 2.10% | 9.00% | 1.84% | 0.40% | 7.20% | 1.00% |

| Hectares Used per Region (1 to 6) for Crops for Energy Purposes and Industry (excl.Starch Production) | | | | | | | |
|---|---|---|---|---|---|---|---|
| **Region** | **Cereals total** | **Wheat, spelt, Einkorn** | **Silage maize** | **Sugar beets** | **Potatoes** | **Winter oilseed rape** | **Pulses** |
| 1 Hesse | 6002 | 2979 | 3955 | 303 | 16 | 3128 | 133 |
| 2 GD Da | 1836 | 1005 | 1053 | 163 | 12 | 826 | 33 |
| 3 GD Gi | 1507 | 693 | 919 | 27 | 2 | 830 | 37 |
| 4 GD Ka | 2659 | 1281 | 1983 | 113 | 3 | 1472 | 63 |
| 5 F MPA | 1433 | 799 | 825 | 135 | 4 | 563 | 24 |
| 6 M-B | 410 | 170 | 224 | 6 | 0 | 159 | 9 |

Additional information on Table A7 and Table S4: In Germany, about 20% of total cropland is used for energy sourcing or industrial purposes, mainly biogas (53%, two-thirds of which is maize), followed by fuel (36%, 74% of which is rapeseed) and ethanol (26%; mainly wheat, rye, sugar beet and corn maize). About 11% of the 20% of land is used for technical purposes (46%), starch (45%), industrial sugar (5%) or colouring plants (4%). About 60% of the total cropland is used for livestock fodder. Only about 20–22% of the crops produced are for human consumption. Specific data for Hesse were not available; thus, we assumed the same shares. In further calculations, these shares were deducted [66,82,83].

**Table A8.** Hectares Available for Direct Human Consumption. Note: The data for Cereals total (region 1: 256,535 ha) is taken from Table A1 Cereals total (region 1: 289,347 ha) minus triticale (region 1: 19,342 ha) and corn maize/corn-cob mix (region 1: 13,470 ha), as both is grown for animal feed.

| Available ha for Direct Human Consumption, Regions 1 to 6 | | | | | | | |
|---|---|---|---|---|---|---|---|
| **Region** | **Cereals Total** | **Sugar Beets** | **Potatoes** | **Rape Seeds** | **Legumes (5%)** | **Vegetables** | **Sum** |
| 1-Hesse | 256,535 | 16,504 | 4421 | 43,204 | 671 | 7494 | 328,829 |
| 2-GD Da | 79,108 | 8845 | 3192 | 11,406 | 165 | 6582 | 109,298 |
| 3-GD Gi | 63,785 | 1486 | 487 | 11,465 | 187 | 150 | 77,560 |
| 4-GD Ka | 113,643 | 6173 | 742 | 20,333 | 319 | 763 | 141,973 |
| 5-F MPA | 61,305 | 7361 | 1014 | 7773 | 123 | 4569 | 82,145 |
| 6-M-B | 16,355 | 313 | 77 | 2200 | 47 | 35 | 19,027 |

**Table A9.** Self-Sufficiency Degrees of Current Consumption Based on Hectares Available for Direct Human Consumption (cf. Table A8).

| Self-Sufficiency Degree of Direct Human Consumption per Crop, Regions 1 to 6 (Current Consumption) | | | | | | | |
|---|---|---|---|---|---|---|---|
| **Region** | **Cereals Total** | **Sugar Beets** | **Potatoes** | **Rape Seeds** | **Legumes (5%)** | **Vegetables** | **Sum** |
| 1-Hesse | 327% | 137% | 50% | 45% | 10% | 31% | 145% |
| 2-GD Da | 158% | 115% | 56% | 19% | 4% | 43% | 75% |
| 3-GD Gi | 487% | 74% | 33% | 71% | 17% | 3.7% | 205% |
| 4-GD Ka | 748% | 264% | 43% | 109% | 24% | 16% | 324% |
| 5-F MPA | 166% | 130% | 24% | 17% | 4% | 41% | 77% |
| 6-M-B | 527% | 66% | 22% | 58% | 18% | 3.7% | 213% |

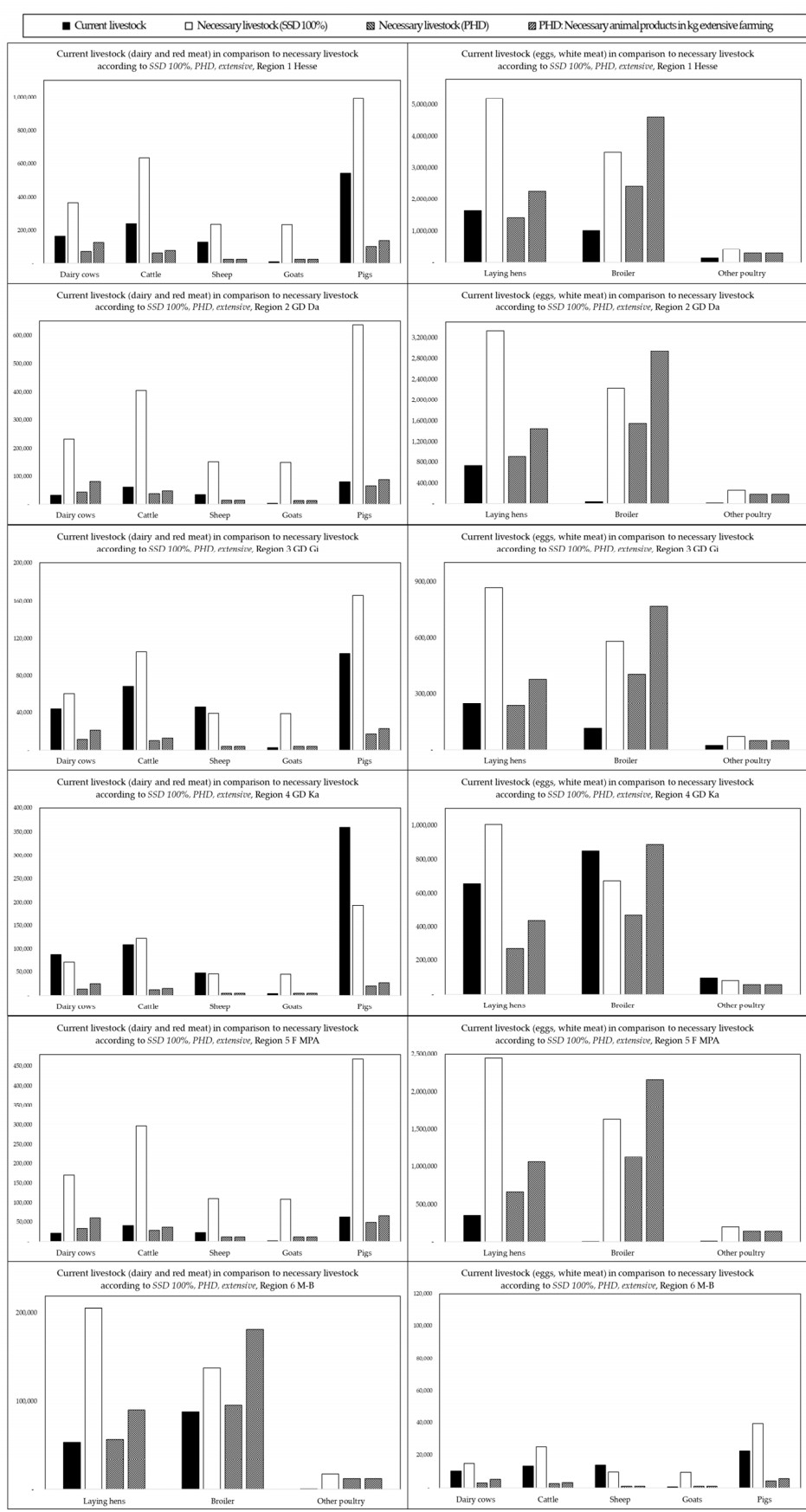

**Figure A1.** Current Livestock for Meat, Dairy and Eggs in Comparison to Necessary Livestock According to SSD 100%, PHD and Extensive in Regions 1 to 6.

**Table A10.** Self-Sufficiency Degrees of Consumption Recommended by Planetary Health Diet Based on Hectares Available for Direct Human Consumption (cf. Table A8).

| | Self-Sufficiency Degree of Direct Human Consumption per Crop, Regions 1 to 6 (PHD) | | | | | | |
|---|---|---|---|---|---|---|---|
| Region | Cereals Total | Sugar Beets | Potatoes | Rape Seeds | Legumes (5%) | Vegetables | Sum |
| 1-Hesse | 373% | 457% | 188% | 65% | 1.1% | 36% | 146% |
| 2-GD Da | 180% | 383% | 212% | 27% | 0.4% | 50% | 76% |
| 3-GD Gi | 556% | 246% | 124% | 104% | 1.8% | 4.3% | 206% |
| 4-GD Ka | 855% | 883% | 163% | 159% | 2.6% | 19% | 326% |
| 5-F MPA | 190% | 434% | 92% | 25% | 0.4% | 47% | 78% |
| 6-M-B | 603% | 219% | 83% | 84% | 1.9% | 4.3% | 214% |

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
