# Peer review of "Land Consumption for Current Diets Compared with That for the Planetary Health Diet—How Many People Can Our Land Feed?"

_sustainability, doi:10.3390/su15118675_

Round 1
Reviewer 1 Report
The manuscript entitled “Land Consumption of Current Diets Compared to the Planetary Health Diet—How many People can our Land Feed?” has aim to calculated and compared land consumption of current diets in the region Hesse in Central Germany, with the consumption as recommended by the Planetary Health Diet.
The topic is very important and UpToDate.
The authors provided a large amount of data and calculations according to the statistical report, but based on them there is no adequate solution offered in terms of nutrition, i.e. in the form of shifts in dietary patterns - consumption reduction per capita.
Unfortunately, nothing new was offered in this manuscript, we already know that we consume too much food that we do not have the resources to produce locally and that this is unsustainable for the planet and that we need national policies in this regard. Also, we have addressed the problem of obesity and malnutrition due to trends in dietary patterns.
The manuscript would be very good as a professional paper.
Some of the comments and suggestions for the manuscript:
Abstract:
Line 7 - The way people in many countries eat today – Dietary habits….
Line 8 - but often cannot be fed by local resources. - Can you rephrase this part of the sentence a bit? I understand that it is about the fact that there is not enough fed for the livestock consumed by the population.
Generally, current consumption patterns are generally not sustainable. Therefore, it is necessary to consider how much and what changes should be made in order for our dietary patterns to be sustainable.
Within the introduction, it is necessary to correct the technical writing reference in several places (e.g. Line 56)
Can you place the white space between the two maps in Figure 1? Considering that there are no highlighted major cites nor any landmarks on 1 map, at first it can be confusing and unclear that these are two identical areas.
Table 1. – is there a statistical data or it is some kind of calculation for median energy intake, considering the cited literature it is not clear
Figure 3 - small font and therefore incomprehensible
English proofreading is recommended, some parts are not clear.
Reviewer 2 Report
Please make sure the abstract and introduction are clear including an economic argument as to why you found something rather than simply saying what you have found from your analysis, highlighting its novelty relative to the literature.
The details of the methodology and type of data used in this research should be presented in the abstract.
The research literature has not been fully reviewed. It is recommended to provide more complete research literature.
The theoretical framework and hypothesis for the study are missing and recommended to incorporate them at the end of the literature review
What is the basis for choosing a food group? It is necessary to provide more explanations in this regard.
This article needs a conclusion and policy implication section.
Round 2
Reviewer 1 Report
I would like to thank the authors for their efforts to improve the manuscript.
Some parts of the manuscript are now much and clearer. Some new references have been added and rearrangements have been made to the text. The parts of the manuscript are more clearly connected.
Technical suggestions:
- scheme (line 205-213) should have a title e.g. Fig 1. the figure together with explanation and description of the abbreviation to be clear without additional text.
- In the last revision I ask if it is possible to put a gap (white space) between to maps on the figure 2 (to be clear what is left and what right part)??
Conclusion should be more concrete.
Reviewer 2 Report
Comments were well addressed.
Author Response
As no further feedback was given, we like to thank the reviewer for his/her time.